# Resistance to Gemcitabine in Pancreatic Ductal Adenocarcinoma: A Physiopathologic and Pharmacologic Review

**DOI:** 10.3390/cancers14102486

**Published:** 2022-05-18

**Authors:** Tomas Koltai, Stephan Joel Reshkin, Tiago M. A. Carvalho, Daria Di Molfetta, Maria Raffaella Greco, Khalid Omer Alfarouk, Rosa Angela Cardone

**Affiliations:** 1Via Pier Capponi 6, 50132 Florence, Italy; 2Department of Biosciences, Biotechnologies and Biopharmaceutics, University of Bari, 70126 Bari, Italy; tiago.amaralcarvalho@uniba.it (T.M.A.C.); daria.dimolfetta@uniba.it (D.D.M.); grecoraffaella@hotmail.it (M.R.G.); rosaangela.cardone@uniba.it (R.A.C.); 3Zamzam Research Center, Zamzam University College, Khartoum 11123, Sudan; alfarouk@hala-alfarouk.org; 4Alfarouk Biomedical Research LLC, Temple Terrace, FL 33617, USA

**Keywords:** pancreatic ductal adenocarcinoma, resistance to treatment, gemcitabine, desmoplastic reaction, hydroxyurea, proteasome inhibitors

## Abstract

**Simple Summary:**

PDAC is one of the most malignant tumors and its treatment, whether surgery or chemotherapy, has shown poor results. Resistance to gemcitabine and other chemotherapeutic drugs is an essential factor in this failure. This review analyzes the molecular causes of gemcitabine resistance and discusses the possibilities of new approaches aimed at decreasing, delaying or even reversing chemoresistance in pancreatic cancer.

**Abstract:**

Pancreatic ductal adenocarcinoma (PDAC) is a very aggressive tumor with a poor prognosis and inadequate response to treatment. Many factors contribute to this therapeutic failure: lack of symptoms until the tumor reaches an advanced stage, leading to late diagnosis; early lymphatic and hematic spread; advanced age of patients; important development of a pro-tumoral and hyperfibrotic stroma; high genetic and metabolic heterogeneity; poor vascular supply; a highly acidic matrix; extreme hypoxia; and early development of resistance to the available therapeutic options. In most cases, the disease is silent for a long time, andwhen it does become symptomatic, it is too late for ablative surgery; this is one of the major reasons explaining the short survival associated with the disease. Even when surgery is possible, relapsesare frequent, andthe causes of this devastating picture are the low efficacy ofand early resistance to all known chemotherapeutic treatments. Thus, it is imperative to analyze the roots of this resistance in order to improve the benefits of therapy. PDAC chemoresistance is the final product of different, but to some extent, interconnected factors. Surgery, being the most adequate treatment for pancreatic cancer and the only one that in a few selected cases can achieve longer survival, is only possible in less than 20% of patients. Thus, the treatment burden relies on chemotherapy in mostcases. While the FOLFIRINOX scheme has a slightly longer overall survival, it also produces many more adverse eventsso that gemcitabine is still considered the first choice for treatment, especially in combination with other compounds/agents. This review discusses the multiple causes of gemcitabine resistance in PDAC.

## 1. Introduction

Pancreatic cancer is the fourth leading cause of cancer-related mortality in the West, and it is projected to be in the second place in the United States by 2030 [1,2]. Pancreatic ductal adenocarcinoma (PDAC) accounts for 90% of pancreatic cancers [3], and its mortality has been constantly increasing over the past 10 years [4]. Approximately 1.7 percent of men and women will be diagnosed with pancreatic cancer at some point during their lifetime [5]. The major risk factors associated with PDAC are age, alcohol consumption, chronic pancreatitis, diabetes, obesity, family history, and tobacco use [6,7,8,9]. With the exception of smoking, which has been decreasing, all the other risk factors are on an upswing. While the relationship between PDAC and diabetes is well known [10,11,12], no causality has been clearly identified. Insulin seems to increase risk [13], while metformin decreases it [14,15]. The best proof of the ominous prognosis of this disease is the fact that incidence and mortality are very similar, meaning that almost all patients with pancreatic cancer die fromit. The average age at diagnosis is between 70 and 75 years.

Despite significant breakthroughs in cancer research, PDAC still has high mortalityand is one of the most chemoresistant cancers. The most prominent of the many factors that contribute to this very poor outcome are: heterogeneity of genetic mutations that accumulate with disease progression [16];the dense stromal environment [17]; late diagnosis [18]; age; poor vascularity; extreme hypoxia and extracellular acidosis; early metastasis; lack of initial symptoms; and high incidence of chronic pancreatitis. Early micrometastatic disease is a fundamental issue leading to poor prognosis [19]. Furthermore, Rhim et al. [20] found that malignant cells invaded and entered the circulation very early, even before a tumor could be detected by rigorous histologic analysis.

Approximately 85–90% of tumors are non-resectable at the time of diagnosis [21]. Most patients are in an advanced stage at the time of treatment and the median survival is less thanone year [22] when ablative surgery is not possible. With successful surgery, the overall survival is approximately 20 to 30 months. The recurrence rate among patients undergoing a full-tumor resection is very high, and less than half of them are able to complete postoperative adjuvant chemotherapy [23]. Unfortunately, thesestatistics for pancreatic cancer outcomes have changed little in the last 20 years, meaning that the disease has scarcely benefited from recent advances in oncology. However, according to some very recent vital statistics, there was a tiny increase in overall survival with very aggressive treatment schemes, but the quality of life with those treatments is usually disappointing. There is one exception to these poor results: the CONKO-001 trial showed that *“Among patients with macroscopic complete removal of pancreatic cancer, the use of adjuvant gemcitabine for 6 months compared with observation alone resulted in increased overall survival as well as disease-free survival”* [24]. 

The situation is different in unresectable cancers; although several publications reported some benefits, they are notoriously ineffective in prolonging survival and improving quality of life [25]. Despite improvements in the approaches for detecting and managing pancreatic cancer, the five-year survival rate barely reached9% in 2020 [26]. Some risk factors can be reduced, such as smoking, obesity, or alcoholism, while others, such as age above 50 years [27], the presence of BRCA2 mutations, Lynch syndrome, and Peutz Jeggers syndrome [6] cannot be mitigated [28]. Diabetes and chronic pancreatitis occupy an intermediate position.

Frequent and early metastasis and retroperitoneal infiltration preclude surgery [29]. Current chemotherapy, mainly based on gemcitabine as the gold standard (with or without nab-paclitaxel [30], or with cisplatin), has shown some very modest improvements, but they are far from achieving acceptable results [31]. The FOLFIRINOX scheme (folinic acid/5-FU/irinotecan/oxaliplatin) has been found somewhat more effective, but at the price of high toxicity [32,33]. Tyrosine-kinase inhibitors, such as erlotinib, are effective in the laboratory but not in the clinical setting [34]. Antiangiogenic treatments have also failed clinical testing [35]. This is logical because antiangiogenesis increases hypoxia and the already existing poor vascular supply in PDAC, thus further decreasing possibilities for other chemotherapeutics to reach the tumor. Furthermore, PDAC uses vasculogenic mimicry very actively and this is not affected by antiangiogenics [36,37,38].

Neoadjuvant chemotherapy has slightly increased the proportion of patients that can undergo ablative surgery, particularly in those cases considered to be borderline resectable tumors [39,40,41,42]. Survival benefits have also been observed with neoadjuvant chemotherapy before chemoradiation [43].

PDAC frequently uses autophagy as a resource, even without chemotherapy [44]. We are unable to determine the reasons leading to this autophagic behavior. We can speculate that the poor vascular supply produces a shortage of nutrients that can be handled by reshuffling biomolecules from unnecessary metabolites [45].

It is also quite possible that autophagy in PDAC plays an important role in cytotoxicity escape [46]. It has been shown that autophagy may function as a tumor suppressor. However, in PDAC it seems to function in favor of tumor resistance [47]. 

Most cases of pancreatic cancer are produced by sporadic genetic mutations; however, there are a few that are related to familial and hereditary factors [48,49,50,51,52,53]. Patients with these germline mutations should beexamined with the only clinical available method that can detect early disease before clinical manifestations: abdominal CT scan [54].

Late diagnosis as well as chemo- and radio-resistance are probably the main causes of treatment failure. This review focuses on chemoresistance to gemcitabine, which is the first-line chemotherapeutic treatment of choice.

### Initiation and Progression of PDAC

Many possible causative factors have been identified as initiators of this tumor type, including high plasticity of acinar cells (dedifferentiation into pluripotential cells known as acinar-ductal metaplasia), intra-and peri-tumoral inflammation (including acute and chronic pancreatitis), immunosurveillance failure, KRAS mutation, hyperglycemia, highly variable extracellular pH in acid–base transporting epithelia, ROS regulation by TIGAR, exosomes, nicotine, nicotinic acetyl choline receptors, ERstress protein AGR2,autophagy, and many others [55,56,57,58,59,60,61,62,63,64,65,66,67,68,69,70,71,72,73,74,75,76,77,78,79,80,81,82,83,84,85,86,87,88,89,90,91,92,93,94,95,96,97,98,99,100,101,102,103,104,105,106,107,108,109,110,111,112,113,114,115,116,117,118,119,120,121,122,123,124,125,126,127,128]. The large quantity of proposed tumor initiators leads us to believe that many authors include many of the mechanisms that participate in tumor progression as tumor initiators rather than initators themselves.

Although the precise initiator of PDAC remains elusive, the following facts are clearly known:(a)There are germline and somatic mutations, such as KRAS, p53, p16 and SMAD4,that predispose to PDAC [129,130,131,132,133,134,135,136,137,138,139,140,141,142,143,144,145,146,147,148,149];(b)KRAS mutation and activation represent a critical factor in initiation [150,151];(c)Pancreatica cancer originates from acinar and ductal cells [152];(d)Progression from normal cells into invasive ductal adenocarcinoma is the product of multiple mutations [153];(e)Inflammation undoubtedly plays a role in both initiation and progression;(f)We know more about progression than about initiation;it has been established that invasive pancreatic adenocarcinoma is the result of the clonal evolution of severe ductal dysplasia [154].

## 2. Causes of Resistance to Chemotherapy in PDAC

### 2.1. Multidrug Resistance

As in many other tumors, multidrug resistance (MDR) is a frequentoccurrence in pancreatic cancer [155,156]. Surprisingly, Suwa et al. [157] reported that P-gp/MDR1 expression in untreated patients carried a better prognosis. They also reported that the expression of this protein was higher in 73% of the 103 untreated pancreatic tumors they tested. The three main MDR proteins, namely MDR1, MRP, andBCRP were found to be increased in PDAC, both with and without treatment [156,158]. In this regard, PDAC shows MDR characteristics similar to other tumors.

### 2.2. Desmoplastic Stromal Reaction

PDAC is histopathologically characterized by desmoplasia, consisting of a densely packed fibrotic extracellular matrix (ECM) [159,160].

The desmoplastic reaction surrounds pancreatic tumors and represents an iron shield that is able to impede therapeutic interventions of different natures.

The desmoplastic reaction is not only the hallmark of pancreatic cancer but also of chronic pancreatitis. Components of the desmoplastic reaction are collagens, fibronectin, and hyaluronan, abundantly secreted by the specialized stromal myofibroblastic cells known as stellate cells [161]. Cancer-specific alterations in ECM architecture have gained significant attention with the increased recognition that this abnormality has therapeutic consequences through the following:Its effects on tumor mechanics [162];Changes in cancer cell migration/invasion [163,164,165];Decreased drug penetration into the tumor [166].

The existence of this dense tumor microenvironment (TME) may be the main reason that therapies specifically targeting only cancer-associated molecular pathways have not shownbetter results [155].

Originally, the cancer cell was considered the main culprit of this peculiar ECM production [167]. Evidence has shown that this is not so. There are specialized cells, i.e.,stellate cells (SCs), that develop this peculiar ECM. In the normal pancreas, SCs are in a quiescent stage surrounding the pancreatic acini. In chronic pancreatitis and when malignancy develops, they become active participants in the process. A symbiotic relationship between malignancy and SCs was proposed [168,169]; Vonlaufen called it an “unholy alliance” [170].

SCs are modified fibroblasts that adopt a myofibroblastic aspect.

For the researcher, the desmoplastic reaction represents a serious obstacle for studying isolated PDAC cells in vitro due to the fact that they lack a similar stroma as found in vivo [171]. Thus a co-culture of cancer cells and myofibroblasts from pancreatic stroma is a necessary step for basic research. 

Usually, the tumor microenvironment [172] (TME) of PDAC is characterized by abundant stroma, hypoxia, deficient blood supply, and elevated immunosuppression [173]. Studies have shown that the TME, including cancer-associated fibroblasts (CAFs), stellate cells, tumor associated macrophages (TAMs), and diverse immune cells and the cytokines they release, are involved in the control of the proliferation, metastasis, chemoresistance, and disruption of immunosurveillance of pancreatic cancer [174]. Factors associated with TME, such as cell plasticity, tumor heterogeneity, composition of the tumor stroma, epithelial-to-mesenchymal transition (EMT), reprogramming metabolism, acidic extracellular pH (pH_e_), and hypoxia can heavily impact treatment outcomes. Therefore, finding new therapeutic targets within PDAC’s TME is a research goal to pursue

Initially, the role of this phenomenon was overlooked; however, various studies have since demonstrated that, during PDAC development, the cancer cells expend a large amount of energy in promoting the recruitment, proliferation and activation of fibroblasts. Consequent to their activation, stellate cells are able to deposit ECM and secrete several types of factors that strongly affect the behavior of cancer cells [168,175,176]. Indeed, pharmacologic blocking of the desmoplastic reaction, in combination with chemotherapy, showed better results in inhibiting PDAC progression than chemotherapy alone, thus highlighting desmoplasia as a likely therapeutic target in pancreatic cancer [177,178,179,180].

The histological manifestations of desmoplasia can be divided into two categories:Considerable overproduction of ECM proteins;Extensive proliferation of myofibroblast-like cells (stellate cells) [181,182].

Therefore, the resulting dense and fibrous mesenchymal tissue is comprised of both cellular and non-cellular components. In this section, we focus on the non-cellular components.

The non-cellular components of desmoplasia include multiple ECM proteins, namely, collagen types I, III, IV, and XV, fibronectin, laminin, hyaluronan, as well as the glycoprotein osteonectin [183,184,185]. Desmoplastic progression is the result of several intercellular and intracellular signaling processes. Many reports have shown that transforming growth factor beta (TGFβ), basic fibroblast growth factor, connective tissue growth factor (CTGF), and interleukin-1β are able to stimulate ECM production and, consequently, desmoplastic progression [186,187,188,189,190]. The ECM components can also be divided into two categories: the fibrous proteins, such as collagen, and the polysaccharide chain glycosaminoglycans (GAGs), such as hyaluronan [191,192,193,194]. In the normal pancreas, GAGs play a structural role maintaining compressive forces on the tissue, whereas the fibrous proteins act by supporting the tensile forces on the tissue [195]. In the diseased pancreas, the marked overproduction of ECM constituents can be viewed as the failed resolution of wound healing, leading to fibrosis. Increased expression of collagen types I, III, and IV has been reported through immunohistochemical analyses of pancreatic cancer tissues [196,197]. This over-expression is directly linked to TGFβ/Smad signaling and is the product of fibroblast activity [198]. Remarkably, in pancreatic cancer, this up-regulation of collagen decreases tissue elasticity and increases interstitial fluid pressure, resulting in reduced drug perfusion [199]. Furthermore, collagen production is one of the mechanisms malignant cells use to survive the harsh acidity of the microenvironment [200]. The protein-free GAG, hyaluronan, is also an important component of the ECM, contributing to tissue rigidity and thereby decreasing elasticity [201], and its accumulation within damaged tissue is the product of increased secretion by activated fibroblasts in pancreatic cancer [202]. This ECM component continues to interact with water molecules, preserving tissue hydration in the normal pancreas and creating interstitial hypertension in cancer [203]. Lack of adequate lymphatic drainage, associated with hyaluronan-originated interstitial hypertension, is the perfect formula tohinderadequate delivery of chemotherapeutic drugs to the tumor core [204,205,206]. If we were asked to single out one specific factor of PDAC chemoresistance, our choice would be hyaluronan’s hygroscopic abilities (Figure 1).

#### Mechanism of Production of the Desmoplastic Stroma

The desmoplastic reaction is an inflammatory disorder characterized by fibrogenesis and deposition of extracellular matrix. Although the exact mechanism used for generating desmoplasia is not fully known, based on evidence and some speculation, we propose the following steps: the process is initiated by (i) leucocyte infiltration that (ii) produces cytokines that (iii) induce fibroblastic proliferation that (iv) produces and deposits extracellular matrix [207].

The pathogenesis of the disorder is basically the same in different tissues; therefore, we may consider that the desmoplastic reaction in PDAC is not fundamentally different from what happens in other tumors and inflammatory desmoplastic responses. 

In PDAC the primary offender that ignites the inflammatory process is probably the release of pancreatic enzymes from necrotic tumor cells, creating a “micro-pancreatitis”. In 1997, regarding acute pancreatitis, Kingsnorth wrote: “*Disruption of the acinar cell propagates a macrophage derived cytokine response*” [208]. Interestingly, all the cytokines acting in acute pancreatitis are also found as a cause of the desmoplastic reaction, namely tumor necrosis factor (TNF), platelet activating factor, IL-1, IL-6, IL-8, and IL-10 as the main players.

Interestingly, in chronic pancreatitis pancreatic stellate cells respond to cytokine stimulation [209] as follows: Stellate cell proliferation is stimulated by TNF-α and inhibited by IL-6; IL-1 and IL-10 had no effect on stellate cells proliferation;Collagen synthesis is stimulated by TNF-α and IL-10, while inhibited byIL-6, and unaltered by IL-1.

Therefore, in chronic pancreatitis, the production of a fibrotic matrix is mainly related to TNF-α stimulation of stellate cells. Chronic pancreatitis develops a fibrotic matrix [210] which is quite similar to desmoplastic PDAC and produced by the oxidative stress and cytokines acting on stellate cells. Binkley et al. [211] found that PDAC and chronic pancreatitis stellate cells overexpressed a set of 107 shared genes, showing a possible common mechanism in both cases. This shared characteristic of desmoplasia in PDAC and chronic pancreatitis also explainswhy chronic pancreatitis is a major risk factor for pancreatic cancer [212] (Figure 1).

**Figure 1 cancers-14-02486-f001:**
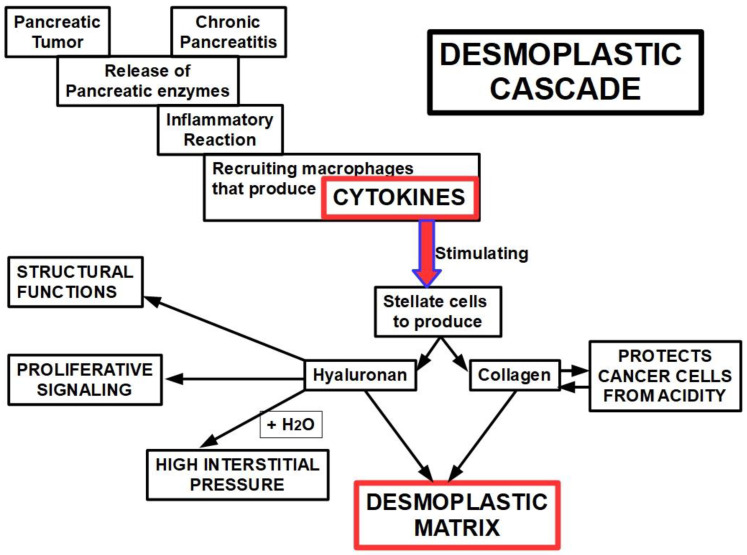
Diagram showing the possible mechanism leading to the desmoplastic reaction. It can betriggered by a pancreatic tumor or chronic pancreatitis. The physiopathology of the process seems similar in pancreatic cancer and chronic pancreatitis. Furthermore, non-active stellate cells can be activated through pancreatic injury, thereby becoming the multi-stellate cells that express alpha-smooth muscle actin (that is the reason they are considered myofibroblasts) [213].

The two issues described above, MDR proteins and desmoplastic stroma, are the core ofPDAC resistance to chemotherapy. 

## 3. Gemcitabine

Gemcitabine was introduced in pancreatic cancer treatment in 1997, after Burris et al. published their report [214]. This was a randomized clinical trialof 126 patients with advanced pancreatic cancer. They found that gemcitabine achieved better results than fluorouracil (5-FU) regarding a modest overall survival improvement and pain control. The mean survival only improved by one month (5.65 for gemcitabine vs. 4.41 for 5FU), but improvements in pain and the Karnofsky index were significant (23.8% for gemcitabine vs. 4.8% for 5FU).

Gemcitabine is still the standard-of-care chemotherapeutic drug for PDAC [215,216]. However, the response rate is quite low (around 30%), and even lower in advanced cases [217,218].

It improves average survival by two to three months [219], a really poor result. Chemoresistance develops rapidly [220] and is therefore the main limiting factor of the drug.

Gemcitabine is used as monotherapy or in combination with other chemotherapeutic drugs [221]. Results in combinatorial treatments are slightly better than monotherapy, however the high toxicity involved in combinatorial schemes led to its being used alone in many cases.

### 3.1. Chemistry

Gemcitabine isa2′, 2′-difluoro 2′ deoxycytidine, a nucleoside useful in the treatment of many different cancers. Figure 2 shows the difference between the deoxycytidine nucleoside that forms part of DNA and gemcitabine, which incorporates two atoms of fluorine. The DNA-synthesizing process is unable to distinguish between the two molecules, and thus incorporates 2-deoxicytidine and gemcitabine indiscriminately. 

### 3.2. Mechanism of Action

In order to study gemcitabine mechanism of action, we must analyze three different steps of activation and one step of inactivation:Drug access to the cell;Intracellular activation;Effects on DNA synthesis;Intracellular inactivation.

#### 3.2.1. Drug Access to the Cell

After the circulatory system delivers the drug to the tumor, gemcitabine encounters some serious problems. The first is the tumor’s decreased vascular supply, and the second, of capital importance, is the dense stroma that surrounds the cell, representing a protective barrier.

Gemcitabine is a hydrophilic moiety, thus its diffusion through the hydrophobic cellmembrane is slow and negligible. Gemcitabine requires transporters in order to enter into the cell [223]. There are two groups of transporters:Human equilibrative nuclear transporters (hENTs), which drive gemcitabine along the direction of the concentration gradient;Human nucleoside concentrative transporters (hCNTs) which are actuallyantiporters that extrude sodium while importing nucleosides. The energy obtained from Na+ extrusion allows the transporters to concentrate gemcitabine against the concentration gradient [224].

##### Human Equilibrative Nucleoside Transporter 1 (hENT1) and 2 (hENT2)

The human equilibrative nucleoside transporters 1 and 2 (hENT1, hENT2), coded by genes *SLC29A1* and *SLC28A1*, aretransmembrane glycoproteins [225] that participate in the bidirectional passage of pyrimidine nucleosides of different kinds, including chemotherapeutic nucleosides such as gemcitabine, capecitabine, and 5-FU. This transport occurs following the concentration gradient, which explains its bidirectionality. Therefore, better intracellular drug access should be expected in patients who express or overexpress these proteins. Gemcitabine is a2′,2′-difluorodeoxycytidine, thus a pyrimidine analogue, and is transported by hENT1 and 2.

##### Structure

hENT1 is consists of 11 subunits that span the cell membrane with a NH2 intracellular terminal, while the COOH end is extracellular. The first extracellular loop joining units 1 and 2 is the site of the union withglycosides (Figure 3).

Greenhalf et al. [226] studieddifferencesinoverall survival among patients who underwent ablative surgery, with high and low expression of hENT1 (ESPAC3 trial). Their findings are shown in Table 1.

Based on these results with a large population study (380 patients), they reached the conclusion that gemcitabine should not be used in patients with low hENT1 expression [226].

A systematic review of 10 studies including 855 patients confirmeda statistically significant longer overall survival in patients with high hENT1expression compared to those with low expression [227]. Thesesurvival benefits in patients treated with gemcitabine and with high expression of hENT1 and deoxycytidine kinase were confirmed in other studies [228,229,230,231,232].

Some authors maintain that hENT1 expression has prognostic value in pancreatic cancer patients treated with gemcitabine [233].

Low gemcitabine cellular import by low hENT1 expression can be improved by loading the drug into nanoparticles. Gao et al. [234] used gemcitabine-loaded human serum albumin nanoparticles, improving cytotoxicity in vitro and in vivo. Interestingly, a dietary product, indole-3-carbinol, was found to increase hENT1 expression. Combining this product with gemcitabine further increased this expression [235]. Indole-3-carbinol is an antioxidant found in cruciferous vegetables and is sold as an over-the-counter dietary supplement. It has independent and controversial anti-cancer effects [236].

##### Human Concentrative Nucleoside Transporters 1 and 3 (hCNT1 and 3)

While hENT1 is a bidirectional transporter of nucleosides, hCNTs work in one direction only, thus concentrating nucleosides (including gemcitabine) inside the cell [237]. 

Reduced CNT1 expression has been found to be associated with gemcitabine resistance [238]. Bhutia et al. [239] compared the level of CNT1 mRNA in tumors with adjacent normal pancreatic tissue. In four out of five tumors it was decreased, by 40% on average, while CNT1 protein was decreased 2-fold. When the comparison was made between normal ductal cells and different pancreatic cancer cells, the decrease was between 24- and 30-fold in all the cell lines. There was a clear correlation between CNT1 expression and gemcitabine influx and cytotoxicity. Only gemcitabine-sensitive cells showed transport activity in spite of decreased CNT1. This activity was almostzero in resistant cells.

While sensitive cells showed the transporter in the membrane, resistant cells showed a low but centrally distributed amount. The conclusion is that: CNT1 expression is reduced in almost all pancreatic ductal tumors;In gemcitabine-resistant cancer cells, CNT1 is also concentratedinside the cell instead ofremaining in the membrane, thus becoming unable to act as a transporter.

MicroARNs (miR) modulate CNT1 protein production. The authors [239] identified miRNA-122, miRNA-214, miRNA-339-3p, and miRNA-650 as downregulating CNT1 transport activity.

The hCNT1 protein is degraded by lysosomes and proteasome.Furthermore, MUC4, a mucin produced by pancreatic cells, is able to reduce CNT1expression, thus reducing gemcitabine penetration into the cell [240].

#### 3.2.2. Gemcitabine’s Intracellular Activation

Inside the cell, the first step of its activation consists in phosphorylation by a deoxcytidine kinase. This is a rate-limiting step (Figure 4).

Acquired downregulation of deoxycytidine kinase impedes the first step of gemcitabine’s activation, thus resulting in resistance [241]. Low expression of deoxycytidin kinase entailed a poor prognosis and shorter survival in patients with resectable pancreatic cancers receiving chemotherapy [242]. 

Two more phosphates are then added by other two enzymes: nucleoside monophosphate kinase and nucleoside diphosphate kinase [243] (Figure 5).

#### 3.2.3. Effects on DNA Synthesis

Difluordeoxycytidine triphosphate is incorporated into new DNA, creating an irreparable error that impedes further DNA formation. This results in cell death. Gemcitabine works as a typical antimetabolite.

A low expression or inactivation of deoxycytidine kinase nullifies or significantly lowers gemcitabine’s action. This has been found to be a frequent mechanism of gemcitabine resistance [244].Gemcitabine also inhibits the fundamental enzyme ribonucleotide reductase (RR), which converts cytidinediphosphate (CDP) into deoxycytidindiphosphate (dCDP) [245] (Figure 6). The intracellularly active gemcitabine isdifluoro-deoxycytidine triphosphate; however, inhibitingribonucleotide reductase seems to be the activity of difluoro deoxycytidine diphosphate [246] (Table 2).

Gemcitabine is a powerful inhibitor of RR that leads to the complete loss of one of the two subunits that form RR, which is probably inactivated by alkylation [247].

In summary: After its second intracellular phosphorylation, gemcitabine produces four effects addressed to block the synthetic phase of the cell cycle: It inhibits ribonucleotide reductase, which converts ribose nucleotides into deoxyribose nucleotides and is the enzyme involved in the synthesis of deoxycytidine monophosphate, which after further phosphorylation is incorporated into DNA;As an antimetabolite, gemcitabine in its active form (gemcitabine triphosphate) is incorporated into the DNA chain, impeding the replication process;Gemcitabine is not excision-repair susceptible, thus indirectly inducing apoptosis;It also exerts inhibitory effects on thymidilate synthase.

#### 3.2.4. Intracellular Inactivation

Gemcitabine is catabolized in tissues through cytidine deaminase. This enzyme converts gemcitabine into 2′,2′-difluorodeoxyuridine. This product competes with gemcitabine uptake because it is transported by both nucleoside transporters hENT and hCNT [248]. This shows that cytidine deaminase plays a double role in gemcitabine resistance: one by inactivating the drug and a secondby indirectly decreasing its delivery into the cell.

Interestingly, if difluorouridine extrusion is blocked, it exerts inhibitory effects on cytidine deaminase [249] (Figure 7).

TAMs induce cytidine deaminase expression, thus inactivating the drug and participating in chemoresistance. Chemotherapy, in general, increases colony-stimulating factor-1 (CSF-1), which increases TAMs infiltration [250,251].

**Figure 7 cancers-14-02486-f007:**
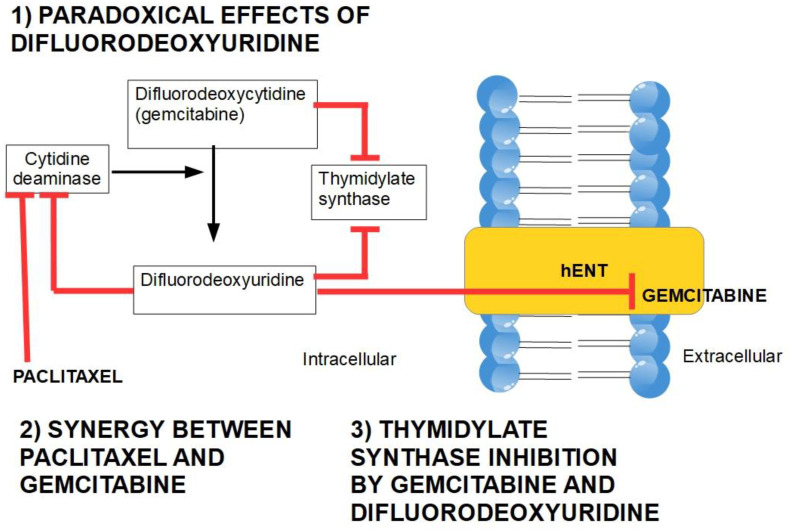
Difluorodeoxyuridine exerts inhibitory effects on cytidine deaminase, thus increasing gemcitabine’s intracellular effects, and it competitively antagonizes gemcitabine intake through hENT. A lower activity of cytidine deaminase is paralleled by a higher cytotoxicity of gemcitabine. This diagram is based on references [248,249,252,253,254,255]. The figure also shows that both gemcitabine and its metabolite difluorodeoxyuridine have the ability to inhibit thymidylate synthase (TS), with further toxicity [256,257]. TS inhibition by 5-FU increased gemcitabine sensitivity [258,259]. Tymidylate synthase inhibition seems to be a valid alternative to gemcitabine in PDAC [260,261].

##### Cytidine Deaminase Inhibitors

There are cytidine deaminase inhibitors in clinical use for the treatment of myelodyspastic syndromes and also for pancreatic cancer. Nab-paclitaxel is a chemotherapeutic taxane that exerts inhibitory effects on cytidine deaminase and is usually associated with gemcitabine, thus reaching synergistic effects [262].

Sohal et al. found that cytidine deaminase activityincreased 10-foldin the plasma of patients with advanced metastatic PDAC compared with patients with resectable tumors, showing that metastases were also important catabolizers of gemcitabine. They used tetrahydrouridine [263] as an inhibitor of the enzyme but did not obtain clinical benefits.

## 4. Mechanisms of Resistance to Gemcitabine

There are multiple mechanisms and participants ingemcitabine resistance. The following factors have been identified:Decrease in deoxycytidine kinase expression or activity [264], thus impeding gemcitabine activation in this rate limiting step (Figure 4) [265,266];Increased expression of Ribonucleotide Reductase isoform M1 [267,268,269], leading to increased production of nucleotides for DNA synthesis;Activation of the PI3K/Akt survival pathway with its anti-apoptotic effect [270];Downregulation of the hypoxia-induced pro-apoptotic gene *BNIP3* [271];Over-expression of focal adhesion kinase (FAK) [272];Over-expression of Src kinase [273,274];c-Met activation: inhibition of c-Met with cabozantinib has overcome gemcitabine resistance and increased its cytotoxicity [275,276,277,278,279];Over-expression of the transcription enhancer high-mobility group A1(HMGA1) [280,281,282];Deoxycytidine release from stellate cells [283];CAF-released exosomal miRNA 106-b [284]: CAFs are intrinsically resistant to gemcitabine and transmit this resistance through exosomes containing miRNA 106-b to cancer cells where they target TP53; For a review on miRNAs in pancreatic cancer, read Slotowinski et al. [285].CAF production ofthe chemokine stromal cell-derived factor 1 (SDF1) is able to activate special AT-rich sequence-binding protein 1 (SATBP1), which intervenes in tumor progression and resistance to gemcitabine [286], as shown in the lower panel of Figure 8;SDF-1α produced by stellate cells and secreted in the stroma has the ability tobind the CXCR4 over-expressed in pancreatic cancer cells, activating a pathway that increases survival, reduces apoptosis, and increasesexpression of IL-6 [287], as shown in Figure 8.TRIM 31 expression by activating NF-kB [288];TGFβ1 also induces gemcitabine resistance [289];Epithelial–mesenchymal transition [290], asthe relationship between EMT and gemcitabine resistance isvery complex:
Gemcitabine-resistant cells acquire EMT phenotype with cancer stem cell characteristics. Notch-2 and Jagged-1 are highly upregulated in these cells [291];Gemcitabine resistance-mediated EMT is in part induced by hypoxia because when HIF-1α is blocked, there is partial reversal of EMT [292]; miR 233 is a contributing factor to gemcitabine resistance-dependent EMT [293];Cells that survive after gemcitabine treatment show increased stemness and EMT markers [294];Gemcitabine-induced EMT sustains chemoresistance [295];Targeting EMTcan overcome resistance [296].

Therefore, based on the above evidence, a circuit like the one shown below may represent the chain of events:**Gemcitabine****→ EMT****→ Resistance to Gemcitabine****→ further EMT**

16.ATP binding cassette (ABC) re-exports cytotoxic compounds in general, including gemcitabine [297,298,299];17.CXCL12-CXCR4 signaling [300,301,302];18.miRNA 320c through SMARCC1 (SMARCC1 is a protein that forms part of the SWI/SNF complex) [303]: This miRNA exerts contradictory actions because it has anti-tumor effects in bladder cancer by downregulating CDK6 [304] and in glioma, where it decreases growth and metastasis [305].It was found to decrease canonical Wnt signaling in joints [306]. Therefore, we can consider miRNA 320can anti-oncogenic miRNA [307] which, however, promotes gemcitabine resistance.19.miRNA 21 and 221 [308,309]: miRNA 21 binds to the 3′-UTR region of the Bcl-2 gene, leading to its over-expression and thus inhibiting apoptosis of pancreatic cancer cells [310]; antisense miRNAs 21 and 221 restored gemcitabine sensitivity and induced cell-cycle arrest and apoptosis [308], while miRNA 200 [311,312] seems to antagonize miRNA21. miRNA 221 is considered a reliable circulating miRNA for diagnostic purposes [313];20.miRNA 155 modulates exosome synthesis and promotes gemcitabine resistance [314]; prolonged treatment with gemcitabine increased miRNA 155 levels, which in turn increased exosomes and expression of anti-apoptotic proteins. The message is carried to the rest of the cells through exosomes.21.miRNA 99a and miRNA100 [315] have been proposed as prognostic markers of gemcitabine resistance;22.miRNA 214 [316,317]:Low expression of miRNA 214 was predictive for improved results of the gemcitabine–vinorelbine association in metastatic esophageal cancer [318];23.miRNA 365 induces gemcitabine resistance by targeting the anti-apoptotic BAX protein and its adaptor protein SHC1 [319]. It also induces the production of survival-related proteins;24.miRNA 210 [320] downregulates Homeobox protein Hox-A9, which increases NF-kB activity and decreases sensitivity to gemcitabine. However, the role of this microRNA is controversial. Amponsah et al. [321] identified miR-210 as a direct suppressor of the multidrug efflux transporter ABCC5; miR210 probably has dualpro- and anti-tumoral effects according to the balance with the oncomucin MUC4. They mutually regulate each other [322];25.miRNA-17-5p is usually overexpressed in PDAC, participating in carcinogenesis and tumor progression [323] and inhibiting Bim expression, thus decreasing apoptosis. Experimental inhibition of miR-17-5 increased sensitivity to gemcitabine [324];

The effects of some miRNAs regarding gemcitabine resistance are still a matter of debate. This is the case of miR-421, which seems to be pro-tumoral, reducing the expression of DPC4/SMAD4 [325] and at the same time increasing gemcitabine efficacy through decreased SPINK1 expression [326]. Furthermore, there is an oleic acid derivative, K73-3, that is able to upregulate miRNA 421 in vitro and in vivo, improving gemcitabine cytotoxicity [327]. Therefore, miRNA 421 should be considered an anti-oncogene agent.

In addition to the miRNAs discussed above, there are others without fully proven inhibitor effects on gemcitabine:26.MUC1 and MUC4 [328,329] (Figure 9): Oncomucins play an important role in gemcitabine resistance that is discussed below. Mucins form a protective envelope surrounding cancer cellsand participate in chemoresistance by impeding drug access to the malignant cells. Their production is usually highly increased in pancreatic cancer. There are two mechanisms involved in oncomucin-induced gemcitabine resistance:
(1)Direct, by MUC1 inhibiting the apoptotic BAX protein and increasing stemness;(2)Indirect, by inducing Her 2 signaling.


Tumor-associated oncomucins have a different glycosylation pattern. MUC1 is less glycosylated than MUC4. MUC1-C, the intracellular portion of MUC1, is a driver for the upregulation of PD-L1. Although this immunoescape was found in triple-negative breast cancer, we can hypothesize that pancreatic cancer’s refractoriness to immune-checkpoint inhibitors may be related to MUC1-C. MUC1-C expression also protects the malignant cells against genotoxic attacks in general;

MUC5AC, a facilitator of migration and invasion, also participates in drug resistance by inhibiting TRAIL death pathways.

27.According to Shukla et al. [344], HIF-1α-dependent highglycolytic flux is the main player in gemcitabine resistance. High glycolytic flux allows for a high cytidine pool that competes with gemcitabine;28.CD44-expressing cells are resistant to gemcitabine. MDR1 is overexpressed in these cells [345]. These CSCs can rebuild the tumor after chemotherapy;29.Tumor heterogeneity: gemcitabine was more effective on cells that were more than 400–500 mμ from the desmoplastic areas [346]. Interestingly, high doses of metformin killed cells closer to the desmoplastic reaction area;30.ROCK2 (Rho associated protein kinase 2) activity is a cause of acquired gemcitabine resistance [347]. The Rhoa/ROCK2 axis promotes migration and metastasis. A pathway has been found in PDAC that shows the long non-coding RNA ZFAS1 inducing metastasis through the Rhoa/ROCK2 axis [348]. ZFAS1 is usually overexpressed in PDAC.ROCK inhibitors sensitize pancreatic CSCs to gemcitabine [349] and also reduce metastasis;31.Constitutive activation of NF-kB [350,351]: IL-1α expression is induced by NF-κB, which in turn increases NF-kB in a positive feedback loop, leading to permanent NF-kB activity [352,353,354,355]. In addition to the classic PI3K/AKT/NF-kB pathway that is fully operative in PDAC, several other pathways that induce gemcitabine resistance through NF-kB activity have been identified [356];

Pancreatic tumors show low miRNA 146-5p expression, impeding regulation of the TRAF6′s 3 UTR segment, thus allowing the pathway shown above. (TRAF6 is the tumor necrosis factor receptor-associated factor 6 that works as an adaptor protein allowing protein–protein interactions) [357].

PARP 14 (Poly ADP-ribose polymerase) is highly expressed in PDAC and is associated with poor prognosis. Silencing PARP 14 reduced resistance to gemcitabine.

Clusterin is a protein associated with chemoresistance to different chemotherapeutics. It was found to be increased in PDAC [358].

In summary, independently of which pathway activates NF-kB, this transcription factor has the ability to eliminate the pro-apoptotic effects of gemcitabine. Blocking NF-kB can, to a certain degree, decrease gemcitabine resistance [359]. 

32.Increased expression of heme-oxygenase-1 (HO-1): PDAC cells show a 6-fold expression of HO-1 compared with normal pancreatic cells. Gemcitabine and/or radiotherapy treatments further increases HO-1 expression. HO-1 knockdown increases sensitivity to both therapies [360];33.Decreased expression of hENT1, the gemcitabine transporter, reduces its intracellular access [361];34.High expression of the polo-like kinase [362]: Downregulation of this kinase decreases resistance [362,363]. Rigosertib, a multikinase inhibitor, has been developed for this purpose. It is being tested in clinical trials [364];35.Decreased glutathione peroxidase 1 induced resistance to gemcitabine: Glutathione peroxidase 1 modulates the AKT/GSK3β/Snail signaling axis in PDAC [365]. Interestingly, gemcitabine is able to induce the expression of glutathione pathway-related genes which are suspected of generating resistance [366];36.Increased expression of Snail [367];37.Increased survivin expression [368]: Emodin, an inhibitor of survivin expression, increases gemcitabine cytotoxicity [369] and similar results can be obtained with small interference RNA (siRNA) [368];38.Decreased intracellular ceramide/sphingosine-1-phosphate [370]: Increased ceramide favors apoptosis, while increased sphingosine-1-phosphate is anti-apoptotic; sphingosine kinase-1 is the enzyme that controls this ratio generating sphingosine-1-kinase, thus exerting anti-apoptotic effects;39.Mutation or deletion of the BRCA2 gene [371];40.Activation of Notch signaling [291,372,373] increases therapeutic resistance: This is related to the acquisition of an epithelial-mesenchymal phenotype (see paragraph 15); downregulation of Notch signaling has a chemosensitizing effect [374]; Notch-induced chemoresistance to gemcitabine is partly the result of Notch’s ability to alter the intrinsic apoptotic pathway [375];41.Hedgehog signaling [376]: chemotherapy activates the Hedgehog pathway [279], and this activation in turn leads to the expression of stem cell markers such as CD44, SOX2, OCT4, Nanog, and drug efflux proteins of the ATP-binding cassette family. Thus, Hedgehog increases stemness and induces a multidrug resistance phenotype [377];42.Cytosolic 5′-nucleotidase 1A over-expression [378]: This enzyme is able to reduce gemcitabine’s intracellular metabolites [379]. The histone deacetylaseinhibitor trichostatin A has been foundto synergize with gemcitabine, increasing its cytotoxicity, and importantly, inhibiting 5′-nucleotidase [380];43.Pancreatic cancer stem cells [381]: Stemness is a key factor in therapeutic failure in most tumors. CSCs do not respond to chemotherapy and are able to replicate the tumor after cytotoxic destruction of sensitive cells. Theactivation of pancreatic cancer stem cells has shown abilities to promote resistance to gemcitabine. Many of the activators are also involved in resistance. (Figure 10);

44.Pancreatic cancer stromal stem cells [402,403];45.Calcyclin-binding protein or Siah-1-interacting protein (CacyBP/SIP) was found to be overexpressed in MDR after gemcitabine treatment. This protein induced P-gp and BCL2 expression reducing apoptosis [404]. In addition, CacyBP/SIP knockdown suppresses proliferation in pancreatic cancer by downregulatingcyclin E and CDK2 and upregulating Rb and p27 [405];46.Soluble V-CAM, produced by pancreatic cancer cells, recruits tumor-associated macrophages (TAMs) [406];47.De novo lipid synthesis [407];48.The extracellular matrix composition: Laminin and collagen type IV-ECM (mimicking an early tumor ECM) protects fromdrug-induced apoptosis compared to a collagen I-rich late-tumor ECM;49.Increased galectin 1 expression in stellate cells [408,409,410,411]: MiRNA 22 was found to reduce the expression of galectin1 in hepatocellular carcinoma [412];50.Autophagy has been shown to be upregulated in PDAC and it plays an important role in resistance tochemotherapy [413,414]. Autophagy is an inducer of gemcitabine resistance and is probably one of the mechanisms that cells use to survive cytotoxic drugs. Gemcitabine’s cytotoxicity is increased when an autophagy-inhibitor is used simultaneously [415]. Pancreatic adenocarcinoma is a very hypoxic tumor, and hypoxia can induce autophagy. Additionally, the expression of high-mobility group box 1 (HMGB1) is an autophagy inducer. Interestingly, gemcitabineupregulates this protein, thus indirectly increasing autophagy [416]. In a preoperative setting, when combining the autophagy inhibitor hydroxychloroquine with gemcitabine, 61% of patients showed CA19.9 marker decrease, improved postoperative, and disease-free survival. These findings were particularly evident in the patients with high levels of the autophagy marker LC3-II [417]. By blocking autophagy, gemcitabine’s cytotoxic effects were increased and stem cell activity reduced [418]. Zeh et al. [419] studied two cohorts of preoperative patients, one receiving nab-placlitaxel, and another group with the same medication plus hydroxychloroquine. They found that the resected pancreas in the hydroxychloroquine group had a greater pathologic response and higher immune activity. However, overall survival and disease- free survival was similar in both cohorts. SNHG14 (small nuclear RNA host gene 14) oncogene expression generates a long non-coding RNA that induces autophagy and resistance to gemcitabine [420]. This LNC-RNA seems to act as an anti-sense against MiRs involved in anti-tumoral activity;

The conclusion is that there is evidence supporting better results with longer overall survival and disease-free survival by adding autophagy inhibitors to gemcitabine in the resectable cases [421]. Evidence in this regard is lacking forinoperable patients.

51.Pancreatic tumor microbiota: There is a clinically important population of bacteria and fungi within the pancreas and biliary tree in patients with PDAC and this population is different from the microbiota found in the normal pancreas [422,423,424]. The bacteria present in PDAC show some specificity [425].Regarding gemcitabine, it was found that intratumoral Gammaproteobacteriahad a role in resistance [426]. Patients that had some surgical or endoscopic procedureon the pancreas and the biliary tree were prone to host pro-resistance bacteria in the pancreas [427], and 5-FU resistance was associated with the presence of Fusobacterium nucleatum in colorectal cancer [428]. Fusobacterium is very abundant in PDAC, so it can be hypothesizedthat it also plays a role in pancreatic chemoresistance. Furthermore, Fusobacterium induces autophagy as part ofits chemoresistance mechanism, another frequent finding in PDAC. Fungi have also been found to be a possible cause of gemcitabine resistance [429];52.Hypoxia is a key factor in the PDAC phenotype, including proliferation, autophagy, progression, metastasis, as well as resistance to treatment in general, and to gemcitabine in particular [430]. The evidence is compelling [431,432,433,434,435,436,437,438]. A simple example shows the importance of this issue. Hypoxia is expressedthrough the hypoxia-inducible factors (transcription factors that modulate over 150 genes). Downregulation of HIF-1α with a newly developed molecule, LW6, inhibited autophagic flux, improved the efficacy of gemcitabine, stopped proliferation, and induced cell death [439]. LW6 is a novelHIF-1inhibitor that decreasesHIF-1αprotein expression [440,441,442];

Hypoxia not only increases resistance to gemcitabine, it also increases gemcitabine-induced stemness [443]. Luo et al. [444] showed that hypoxia induced miRNA 301a which in turn promoted gemcitabine resistance through downregulation of T53, thereby integrating hypoxia, miR, and gemcitabine resistance into one pathway. Figure 11.

53.Increased expression of cytoplasmic ribonucleotide reductase subunit M1 (RRM1) [445]: RR is a multimeric enzyme essential for maintaining a high pool of deoxynucleotides for DNA elongation and also for DNA repair. Gemcitabine-resistant pancreatic cancer cells treated with RRM1 inhibitors showed considerable decrease in resistance [268]. Patients with high RRM1 levels showed a poorer overall survival with gemcitabine treatment compared with low RRM1- expressing patients [446];54.The Hippo pathway is involved in organ size control and tissue homeostasis. It was found that this pathway plays a role in drug resistance [115,447].

Figure 12 presents a summary of mechanisms involved in gemcitabine resistance.

The many and complex different mechanisms involved in gemcitabine resistance would seem to support the nihilist idea that it will be very difficult to solve this problem. However, in a small group of patients, one of us (T.K. unpublished data) found that the iron-chelating agents (through reduction of ribonucleotide reductase) nelfinavir (AKT inhibitor and weak multikinase inhibitor) and fenofibrate (AKT inhibitor) had a limited, albeit positive, effect in delaying resistance. The small number of patients treated does not allowdefinitive conclusionsto be reached. 

## 5. Searching for Possible Solutions

We have presented more than 50 mechanisms proven to play a role in resistance to gemcitabine. Therefore, it is not easy to findone solution that fits all the situations. MDR. For example, Verapamil, a classical P-gp antagonist, and its analogs have been used to block MDR proteins with variable results [448,449], including resistance in PDAC [450,451]. Calcium channel blockers, such as verapamil, were able to decrease pancreatic cancer cell proliferation independently of any effect on MDR proteins [452].

Desmoplastic stromal reaction: Desmoplasia represents a formidable barrier that prevents chemotherapeutic drugs from accessing malignant cells. It is the product of an inflammatory phenotype induced by cytokines secreted by stellate cells and other tumor associated cells. Its main characteristic is the production of a collagen-rich microenvironment that surrounds groups of neoplastic cells.

Multiple possible solutions have been explored. The following drugs have shown some results in this endeavor:Aspirin is able to reduce the inflammatory context that induces desmoplasia in PDAC and increases gemcitabine cytotoxicity [453];Metformin [346,454,455,456,457,458] downregulates TGF-β, suppressing the fibrogenic activity of stellate cells [459] anddecreasing the expression of sonic hedgehog [460]. However, Zechner et al. [461] found that metformin reduced the cytotoxic effects of gemcitabine;4-methylumbelliferone(4MU) is ahydroxycoumarinthat inhibits hyaluronan synthase and decreases the production of hyaluronan. Since hyaluronan is a very hygroscopic compound, its overproduction in the tumor stroma increasesinterstitial pressure, thus impeding drug access to malignant cells. 4MU has been shown to increase not only gemcitabine’s cytotoxicity [462] but 5-FU’s as well [463]. Independently of its effects on hyaluronan reduction, 4MU is able to reduce proliferation in pancreatic cancer [464] and other tumors [465];Pirfenidone is an FDA approved drug for the treatment of idiopathic pulmonary fibrosis. It decreases TGF-β [466] and TNF-α [467] expressions, thus reducing collagen synthesis [468] and collagen fibrils assembly [469]. It has not been tested in pancreatic cancer;Nintedanib is a tyrosine kinase inhibitor approved in Europe for the treatment of idiopathic pulmonary fibrosis. ItinhibitsPDGFR α and β, FGFR, VEGFR, Src, and Lck (lymphocytic tyrosine kinase). These inhibitions block the cascades of signals driving the remodeling of fibrotic tissues [470,471]. It is also a powerful antiangiogenic [472]. Importantly, it has been tested against pancreatic carcinoma with and without gemcitabine association. Nintedanib inhibited proliferation of cells of different lines of PDAC and increased gemcitabine’s cytotoxicity [473]. It decreased the metastatic burden in an experimental model of PDAC [474,475]. Nintedanib is undergoing clinical trials for many solid tumors. NCT02902484 is a phase I, II trial of nintedanib in PDAC as monotherapyand nintedanib followed by gemcitabine plus nab-paclitaxel;All transretinoic acid (ATRA) is able to target stellate cells. A phase I clinical trial with ATRA, gemcitabine, and nab-paclitaxel determined the safety of the association and now a phase II trial is in progress [476]. Independent from ATRAs anti-fibrosis effect, it has direct cytotoxity on PDAC cells. ATRA has shown anti-fibrotic effects in lung after prolonged administration of bleomycin or radiation [477] by downregulatingTGF-β1/Smad3 signaling [478] and also by inhibitingthe IL-6/IL-6R pathway [479]. Similar anti-fibrotic effects were found in the liver, intestine, and kidney. McCarroll et al. [480] found that vitamin A and its derivatives inhibited the activation of pancreatic stellate cells. Retinol and its derivatives ATRA and 9-RA inhibited cell proliferation, and production of collagen I, fibronectin, and laminin in alcohol-induced pancreatic fibrosis. Hisamori et al. [481] found that ATRA downregulated the production of TGF-β1, interleukin-6 (IL-6), collagen, nuclear factor-κB p65, and p38 mitogen-activated protein kinase (p38MAPK) in human hepatic stellate cells. In spite of the objections raised by some authorsthat ATRA can produce exactly the opposite effect, i.e., increase fibrosis [482], the evidence backing ATRA’s anti-fibrotic abilities is strong [483,484,485];Other drugs for targeting desmoplasia include: angiotensin II receptor inhibitors such as candesartan and olmesartan, curcumin, resveratrol, HDAC inhibitors [486,487], and statins. Most of these drugs show anti-fibrotic effects in the liver but have not been tested in pancreatic cancer.

## 6. Main Signaling Pathways Involved in Gemcitabine Resistance

### 6.1. PI3K/AKT and MAPkinase Pathways

PI3K/AKT and MAPkinasepathways areclassically active transduction systems in almost all tumors, including pancreatic cancers. While MAPkinases are mainly pro-proliferative, PI3K/AKT is related to protein synthesis, inhibition of apoptosis, pro-survival, and NF-kB-mediated inflammatory pro-tumoral effects. Thus, these pathways, and particularly PI3K/AKT, play an important role in resistance to chemotherapy [488,489,490,491,492]. Furthermore, gemcitabine’s cytotoxicity can be increased by modulatingthe PI3K/AKT pathway. In this regard, Wei et al. [493] associated evodiamine with gemcitabine, increasing apoptosis in vitro and in vivo. Evodiamine is an over the counter nutraceutical that downregulates PI3K, AKT, and NF-kB [494].

Stellate cells produce miRNA 5703, which upregulates the PI3K pathway in pancreatic cancer cells via exosomes [495]. Here, we have a clear example of the cross-talk between stroma and cancer through exosomes and miRNA as the messenger delivering a pro-tumoral signal that drives gemcitabine resistance. 

There is strong evidence showing that inhibitingthe PI3K pathway sensitizes PDAC to gemcitabine’s effects [496,497,498,499].

### 6.2. CD44

CD44 is a cell surface protein that is activated by hyaluronan binding, which initiates a pro-tumoral signaling cascade, including resistance to gemcitabine. Signaling born from the intracellular portion of CD44 activates Ras, MAPK, PI3K [500], and RUNX2-RANKL pathways [501].CD44 is a marker of cancer stem cells (CSCs) and regulates stemness [502,503,504,505]. It has been shown that CD44 plays an important role in favoring chemoresistance [506]. The mechanism is probably through the hyaluronan–CD44 signaling pathway [507]. Importantly, CD44 is usually overexpressed on the membrane of PDAC cells [508] and contributes to gemcitabine resistance [509,510]. Furthermore, gemcitabine induces CD44 expression on the cell surface [511]. Based on this evidence, it is clear that the hyaluronan–CD44 axis needs to be blocked in order to decrease or delay gemcitabine resistance.

Bromelain is a nutraceutical with anti-inflammatory properties thatcan reduce CD44cell surface presence [512] and in general modulates CD44’s expression [513].

### 6.3. IL-6/IL-6R/STAT3 Axis

IL-6 promotes the accumulation of myeloid-derived suppressor cells through the IL-6/IL-6R/STAT3 signaling pathway [514] and intervenes in resistance to gemcitabine [515]. IL-6 gene knockdown sensitized pancreatic cancer cells to gemcitabine [516]. In cholangiocarcinoma cells, it was found that gemcitabine upregulated IL-6 and IL-8 [517]. (Figure 13). We mentioned above that TAMs exert chemoresistance through cytidine deaminase upregulation. Furthermore, stellate cells are also involved in the production/secretion of IL-6 [518], thus the tumor microenvironment is rich in IL-6.

Tocilizumab is a monoclonal antibody directed against the IL-6 axis. It has been shown that tocilizumab prevents STAT 3 activation in pancreatic cancer [519]. A clinical trial (NCT02866383) is underway to determine if adding tocilizumab to gemcitabine improves outcomes [520].Independently of its role as a possible anti-resistance compound, tocilizumab has direct effects on the cancer by reducing tumor growth and recurrences in xenograft models of pancreatic cancer [521] and was recently reviewed by Sunami et al. [522].

Bazedoxifene (BDF) is an indole derivative acting as a selective estrogen-receptor modulator (SERM) and selective estrogen-receptor degrader (SERD) with mixed agonist and antagonist actions on the estrogen receptor (ER) according to tissue specificity. Interestingly, it is ableto downregulate the IL-6 pathway. This pathway has been found to be active in pancreatic cancer and bazedoxifene has been proposed as part of the treatment as it seems to downregulate the IL6/PG130/STAT3 pathway [523,524,525,526]. In pancreatic cancer, the evidence indicates that the anti-cancer mechanism is independent of BDF’s hormonaleffects on the ERα. The IL6-GP130-STAT3 signaling axis seems to be an important tumor driver in many cancers [527,528,529,530,531], including pancreatic. BDFis able to disrupt this axis by interfering with the IL6R-GP130 relationship, thus blocking GP130 signaling. 

The IL-6 pathway is shown in Figure 14.

## 7. Discussion

Late diagnosis and early metastasis, are at the core of the poor therapeutic results in PDAC, and have not changed substantially in the last 50 years. Pancreatic cancer has some characteristic features that contribute to this failure, such as stromal desmoplasia, low vascularization, and severe hypoxia. These characteristics synergistically contribute to therapeutic resistance.

Cancer chemoresistance in general, and resistance to gemcitabine in pancreatic tumors in particular, developsthrough multiple mechanisms. Essentially, they originate from:Structural barriers to drug absorption such as desmoplastic stroma or low vascular supply to the tumor or;Biological mechanisms and factors within the tumor itself, includinglow expression of drug importers, increased expression of exporters, increased expression of enzymes involved in drug catabolism, increased autophagy, and anti-apoptotic proteins.

When the more than 50 identified mechanisms of gemcitabine resistance are analyzed in depth, it becomes clear that, actually, they can be simplified into sevenkey groups:MDR and MDR inducers (drug extruders);Desmoplasia and desmoplasiainducers (physical barrier) [532];Non NF-kB-related anti-apoptotics (increased expression of anti-apoptotic activity);NF-kB-mediated anti-apoptosis (increased expression of anti-apoptotic activity);Low levels of hENT1 (decreased expression of drug import transporters) and/or accelerated gemcitabine deamination;Low (acidic) extracellular pH and increased exosome release;Oncomucins.

Oncomucins, such as MUC1 (CD227), increase the expression of MDR proteins by acting astranscription factors in the promoter region of the *ABCC1* gene [533].

MUC4, on the other hand, seems to inhibit apoptosis by indirectly inactivatingthe anti-apoptotic protein Bad [534]. Since MUC4 builds up slowly while the tumor progresses [535], we can speculate that in advanced tumors it plays a role in intrinsic resistance. In addition, MUC4 has also other protumoral effects such as interacting with and stabilizing the Her2 receptor [536] and fibroblast growth factor receptor 1 (FGFR1) [328]. Both oncomucins indirectly activate the AKT pro-survival and anti-apoptotic pathways. (See Figure 9). There is evidence that the link between chronic pancreatitis and PDAC may be MUC1-C [537], which promotes signaling pathways found in pancreatic cancer and in wound healing as well [538]. For a recent review read Li, et al. [539].

Other important players are as follows:Cytokines are key participants in the creation of a collagen and hyaluronan-rich dense stroma, thushindering gemcitabine’s access to the cell. Many different cytokines converge into inducing IL-6, the major player in the cytokine orchestra that promotes desmoplastic reaction, pro-tumoral, and anti-apoptotic pathways;The pro-inflammatory NF-kB transcription factoris part of many pro-tumoral pathways. PI3K/AKT/NF-kB/Bcl2 is particularly interesting:it acts as a driver pathway in many pancreatic cancers and is a major player in gemcitabine resistance. Downregulating any of the members of the pathway restores sensitivity to chemotherapy [540]. NF-kB is the final molecule towards which many proteins and miRs, such as PARP14 [357], clusterin [358], and miRNA 146A [356], converge in order to induce chemoresistance;Tumor-stroma crosstalk is not only a pivotal fact in PDAC progression and metastasis but also a key component of chemoresistance. In this regard, addressing only MDR proteins is not enough. The tumor and its peculiar stroma must be targeted simultaneously.

In addition, there are many other proteins and pathways that play a role in resistance.

Macrophages: Tumor associated macrophages (TAMs) have a multifaceted relationship with gemcitabine resistance.
(1)TAMs release pyrimidines that compete with- and decrease gemcitabine’s effects [541];(2)Gemcitabine recruits macrophages into the tumors. Furthermore, it induces them to adopt the M2 phenotype that has immunosuppressive, pro-tumoral and drug resistance capabilities [542,543];(3)The proof of concept lies in the fact that depletion of TAMs improves gemcitabine cytotoxic effects [544];
Cancer-associated fibroblasts:Cancer-associated fibroblasts (CAFs) are fibroblasts that have been functionally “sequestered” by the tumor. We can call this “enslavement”. CAFs participate in [545] extracellular matrix (ECM) remodeling;metabolism modulation;energy source for the tumor (lactate shuttle); angiogenesis modulation;production of growth factors, cytokines, and chemokines; collagen production; immunosuppression; and drug resistance;

CAF-mediated drug resistance has many aspects and can be divided into soluble factor-mediated drug resistance and cell adhesion-mediated drug resistance [546].

In the first case, CAFs produce different pro-tumoral compounds, including cytokines, such as TGF-β, TNF-α, IL-1,growth factors, and exosomes, inducing desmoplastic reactions. These effects impede chemotherapy-induced apoptosis.

CAFs decrease CD8+ T lymphocyte’s function and recruit T regulatory cells (Tregs) to the tumor [547].CAFs generate resistance to gemcitabine through the SDF-1/SATB-1 pathway. SDF-1 issecreted by CAFs stimulating malignant progression and gemcitabine resistance in pancreatic cancer (see Figure 4).Other resistance pathways related to CAFs are shown in Figure 15.

Extracellular acidity: While there is no direct evidence that extracellular acidity interferes directly with gemcitabine, it is an important element in immune escape. Chemotherapy works better when there is a competent immune system. Thus, reducing extracellular acidity with simple and non-toxic drugs should represent extra help in most cancer protocols;Administration schedule: Gemcitabine is usually administered in a onceweekly dose of 1000 mg/m^2^ for three weeksfollowed by a one week rest. Then, the cycle is repeated. This scheme is a standard MTD (maximum tolerated dose). However, there are other schemes that may be more effective regarding cytotoxicity. The general idea behind alternative schedules is to obtain maximum efficacy before resistance develops;

Braakhuis et al. [561] showed, in mice, that different administration schemes (every three days with a total of four doses) could achieve tumor eradication without adding more general toxicity. Thus, gemcitabine is a schedule-dependent drug and dose scheduling is of paramount importance in maximizingits anti-tumor efficacy. Cham et al. [562] showed that a metronomic scheme of gemcitabine, with low dose daily administration, and without interruptions achieved a higher reduction of tumor mass compared with standard MTD treatments. In this regard, *“the total dose of gemcitabine administered over 4 weeks in the metronomic group was less than half of that given in the MTD group*”. Interestingly, metronomic treatment improved tumor perfusion and reduced hypoxia. This would result in better access of the drug to the tumor, and we suggest that it could also reduce/delay chemoresistance.

Exosomes: Stromal cells, whether stellate, cancer associated fibroblasts, T-regulatory cells, or macrophages seem to cross-talk with the tumor through cytokines. Another mechanism that has been gaining recognition is inter-cellular communication through exosomes [563,564];

Exosomes are extracellular vesicles released by normal and cancer cells. Exosomes contain proteins, lipids, glycoproteins, DNA, and RNA. There is evidence to supportthat, in addition to being a mechanism to dispose of unnecessary intracellular molecules, they mainly represent an important intercellular communications system in normaland malignant cells and alsoserve a pro-tumoral function in cancer cells. However, there are also exosomes with anti-cancer properties. Cancer cells release a large number of exosomes exchanging information with other neighboring and more distant cells. Stromal cells, are also able to release exosomes that promote tumor growth. Tumor cells can introduce modifications in stromal cells and vice versa, through exosomes. Reducing production and/or release of exosomes has shown a better response to chemotherapeutics anddecreased cancer progression. Furthermore, there are many drugs already in use for other purposes that are able to decrease exosome performance.

We may consider exosomes as the postmen of cells, delivering letters (actually instructions) throughout the organism. What is not very clear is where the central post office is, meaning that exosome regulation is still a matter to be investigated. Many substanceshave been identified as influencing exosome formation and release, but a central coordinator has not. Another important gap in our knowledge is how exosomes “choose” their load and who—or what—is behind exosome modulation. The way exosomes “select” their content is a capital issue that would explain why there are “good” exosomes [565,566] that carry anti-cancer messages and “bad” exosomes transporting pro-tumoral messages [567]. The “good” exosomes are mainly related to improving immunological defenses and anti-tumor immunity [568]. As examples of “bad” exosomes there are those carrying miRNAs such as 122, 105, 135B, 200, 210, 494, and many more, all related to metastasis, angiogenesis, pre-metastatic niche conditioning, or cell growth. They also carry proteins that increase PD-1 activity and drug resistance. Long non-coding RNAs contained in exosomes promote drug resistance and suppress apoptosis. Some exosomes contain oncoproteins like Met [569] and mutated Kras [570]. Tumor cell exosomes were found to contribute to tumor progression by different mechanisms, such as increasing migration, metastasis, niche conditioning, angiogenesis, drug resistance, stemness, and immunosuppression [571,572,573,574,575,576]. Furthermore, tumors actively produceexosomes at a higher rate than normal cells. The amount of exosomes produced by cancer cells is in the range of many millions, and this high output has been attributed to hypoxia [577].

Exosomeshave been found to play a role in PDAC progression and metastasis [578,579,580]. Exosomes also play a role in resistance [581,582] and recruitment of stellate cells [583]. miRNA 210, an indirect inhibitor of gemcitabine effects, is carried by exosomes [584]. Many other oncogenic miRNAs mentioned above are also carried by exosomes [585], thus makingthemvalid targets in order to decrease resistance (Figure 16).

A good example of this exosome-miRNA-gemcitabine resistance relationship is the research by Patel et al. [586]. They found that pancreatic cancer exosomes carried miRNA 155, which decreased the expression of deoxycytidine kinase, a key enzyme in gemcitabine activation.

Exosomes areinvolved in many immunosuppressor effects, such as proliferation of T regulatory cells, apoptosis of cytotoxic T-cells CD8+, inhibition of natural killer (NK) cells, blocking dendritic cell differentiation [573,587,588,589,590]. This may explain the poor results obtained with immune-checkpoint inhibitors in PDAC.

Amiloride, a diuretic that has been in use for over 50 years, decreases exosome production [591], release [592], and uptake [593]. It also diminishes extracellular acidity by inhibiting NHE1. It should be a drug of interest to curtail the stromal–tumor coordination and reduce oncogenic miRNAs release.

Unfortunately, there is no known mechanism to curtail miRNAs activity at the bedside, so that for the time being, amiloride is the best optionfor decreasing the release of some of them.

Indomethacin increases the cytotoxic effects of chemotherapy drugs by blocking exosomal export of drugs [594,595] and, in addition, inhibits NF-kB and COX2. Indomethacin has other anti-cancer effects, such as decreasing cell migration [596], reducing invasion [597], disrupting autophagy [598], decreasing tumor growth, and preventing cachexia [599].

Exosomes can be used to deliver cargo to cancer cells, thus they may be useful for cancer treatment. One method consists in delivering cytotoxic drugs to the tumor [600]. In some cases this delivery method can overcome drug resistance. Tumors can condition their stroma through exosomes, and at the same time stromal cells are able to induce diverse modifications in tumor cells.

Carbonic anhydrase IX is an enzyme that is highly expressed in hypoxic tumors. This is exactly what happens in PDAC, where it is involved in proliferation, necrosis, and angiogenesis, representing a marker of poor prognosis [601]. Interestingly, gemcitabine can induce carbonic anhydrase IX over-expression [602].

These considerations have led us to identify several groups of drugs that can decrease gemcitabine resistance or increase its efficiency.

They are shown in Figure 17.

There are many candidate drugs and pharmaceutical innovations that may participate in this and bring about the so badly needed improvements in therapy and survival. Some of them, such as adding tocilizumab to nab-paclitaxel and/or cisplatin to gemcitabine, are on the brink of being introduced in standard treatment protocols. Others that have been mentioned in this paper are still on the waiting list. Interestingly, such an unsophisticated drug as aspirin or the more complex nintedanib are also part of this long waiting list.

Tocilizumab has been shown to improve PDAC treatment in the laboratory setting [518] and in vivo [521]. In this regard, three clinical trials (NCT02866383, NCT04258150, and NCT02767557) in which tocilizumab is associated with chemotherapeuticsare in progress. Interestingly, tocilizumab was found to have positive effects in experimental acute pancreatitis [605], but at the same time, tocilizumab can cause pancreatitis [606,607].

Analyzing possible complementary treatments to gemcitabine aimed at reducing chemoresistance, we have eliminated some of the drugs mentioned above, such as nintedanib (due to its multiple side-effects). However, it should be regarded as a stand-alone treatment. We have also eliminated curcumin because its bioavailability is very low. The final result leaves us with six drug groups that may enhance gemcitabine’s effects and prevent resistance. See Figure 11.

Silymarin extracts have shown anti-cancer effects in PDAC and also improve sensitivity to gemcitabine treatment [608,609]. Calcitriol was found to increase gemcitabine uptake while reducing the expression of MDR efflux proteins [610].

Samulitis et al. [611] found that highly invasive gemcitabine resistant cells acquired hypersensitivity to class I and II histone deacetylase inhibitors (HDACIs). There is a large body of evidence supporting these drugs for PDAC treatment [612,613,614,615,616,617,618,619,620]. Furthermore, there is also evidence thatHDACIs synergistically increase gemcitabine’s cytotoxicity [621].

As of February 2022, the Clinical Trials page (clinicaltrials.gov) lists 3130 trials for pancreatic cancer. This unusually high number clearly shows the complexity of the subject. Among them, associations of the drugs shown in Table 3 with gemcitabineare or have been under research.

From this shortened list, we canconclude that there is no lack of drugs to associate with gemcitabine; however, no breakthrough success can be established. The most promising therapeutic approach seems to be iNeo-Vac-P01.

Examining the clinical trials (clinicaltrials.gov) for PDAC, we found eight studies with niraparib, a PARP inhibitor; however, none of them were associated with gemcitabine (studies NCT03601923, NCT04409002, NCT03553004, NCT04493060, NCT04764084, NCT03404960, NCT04673448, and NCT05169437). 

Regarding olaparib [638], there are 16 clinical trials but only 1in which the association of gemcitabine with a PARP inhibitoris being tested (NCT00515866), and this is a phase I study, establishing the maximum tolerated dose of the association. No clinical results are recorded. There is another trial of gemcitabine associated with cisplatin and veliparib, another PARP inhibitor (NCT01585805) [639]. This is also a phase I study.

If a pancreatic tumor is found to have mutated DNA repair genes (DDRs), the logical question is: should PARP inhibitors be associated with gemcitabine from the beginning?

This question is validated by the fact that there is strong evidence showing over-expression of DDR genes in pancreatic cancer:Defective DDR pathways are frequently found in inherited and sporadic PDAC [640];Mathews et al. [641] found that tumor-initiating cells repair breaks in DNA faster after they are challenged with gemcitabine;Golan et al. [642] in a review on DDR gene mutations in pancreatic cancer concluded that: “The DDR-deficient subtype of PDAC constitutes an important, clinically relevant, and actionable subset”;ATM, BRCA1, BRCA2, CDKN2A, PALB2, PMS2, BARD1, CHEK2, MUTYH, MSH6, MSH2, MLH1, STK11, andNBN germline mutations were found in a panel of 25 genes tested in 12% of patients with PDAC [643]. These genes are related to DNA repair;Salo-Mullen et al. [644] identified germline mutations in 15% of patients with pancreatic cancer, including BRCA1/2, MSH2 and MLH1, and in patients with early onset of the disease thenumber went up to 28.6%;Approximately 6 to 7% of metastatic PDAC patients treated with conventional protocols have BRCA1/2 mutations, whichrises to 15–20% in Ashkenazy Jewish populations [645,646];Genetic polymorphism of DDR genes increases the risk of pancreatic cancer [647,648];Importantly, gemcitabine is capable of inhibiting the homologous recombination factor RAD51-dependent DNA double-strand break repair [649];Mutations in homologous recombinant genes, such as BRCA1/2 and RAD51, protect cells from gemcitabine’s cytotoxicity [650] (Figure 18).

This evidence shows that gemcitabine would have a reduced cytotoxicity in germline mutated DNA repair genes and the association of gemcitabine with PARP inhibitors would not be synergistic but rather antagonistic.

Hydroxyurea, a chemotherapeutic drug that inhibits ribonuclease reductase, may be an interesting association with gemcitabine, reducing resistance [651]. In 2003, a phase I study of hydroxyurea associated with gemcitabine for the treatment of solid tumors, including PDAC, achieved stable disease in halfthe patients and one partial response (total number of patients = 24) [652].

Interestingly, the Clinicaltrial.gov page shows no trials regarding the hydroxyurea and gemcitabine association. There is only one trial of hydroxyurea in pancreatic cancer but associated with fluorouracil and interferon α (NCT00019474).

According to Minami et al. blocking RRM1 is the more effective way to circumvent gemcitabine resistance, and they found synergistic effects with the gemcitabine–hydroxyurea association [653].

Artemisinin compounds (DHA), a traditional Chinese medicine, has beensuccessfully using the herb *Artesia annua* against malaria since ancient times [654]; its earliest mention datesto 168 BC inprescriptions for 52 kinds of diseases that were found in the Mawangdui Han dynasty tomb.

Nowadays, artemisinins are well established antimalarial agents with an excellent safety profile. Artemisinin-based combination therapies are now recommended by the World Health Organization (WHO) as first-line treatment of uncomplicated falciparum malaria in allregions wherethe disease is endemic [655].

Interestingly, artemisinin derivatives have shown cytotoxic effects on cancer cells, including pancreatic cancer. Furthermore, these drugs are able to reverse multidrug resistance through interactions with P-gp [656,657].

Efferth et al. [658] showed that artemisinin derivatives had important anti-cancer activity against leukemia and colorectal cancer cell lines. There was significantly lower activity against non-small-cell lung cancer cell lines. They found intermediate activity against melanoma, ovarian, prostate, breast, and renal cancer cell lines. Probably the two most important findings in this research were thatcytotoxicity was comparable to standard anti-cancer drugs, andthis activity was not hampered by multidrug resistance to other chemotherapeutics.

In pancreatic cancer cells, DHA produced apoptosis through upregulation of the death receptor [659]. This was observed in vitro and in vivo [660]. NF-KB pathway downregulation is also involved in DHA anti-cancer activity in the pancreas [661] and potentiates gemcitabine’s effects against the tumor [662]. Anti-angiogenesis was also one of the mechanisms described in DHA’s action on the pancreas [663].

Proteasomal inhibition: more than twenty years ago, Bold et al. [664,665] proposed proteasomal inhibitors to sensitize PDAC to gemcitabine. In this regard, there is evidence supporting proteasomal inhibitors for gemcitabine resistance [666,667,668,669,670,671]. Bortezomib, a first generation proteasomal inhibitor, decreased anti-apoptotic Bcl2 expression [672], thus increasing apoptosis and cytotoxicity. Another important effect of proteasomal inhibition is the increased expression ofgrowth arrest, the DNA damage–inducible protein (Gadd153), and the c-Myc antagonist Mad1 [673], which in theory should enhance the anti-tumoral effects of gemcitabine.

There are two clinical trials registered at clinicaltrials.gov involving the association of bortezomib with gemcitabine: NCT00052689 and NCT00620295. There was no overall survival improvement with the association in one trial [674]. The other trial was a phase I study targeting diverse solid tumors, with a reduced number of patients, and it was not specific for PDAC, thus no conclusions can be obtained about the results [675].

Many new proteasomal inhibitors have been identified since bortezomib has been brought into oncological practice.

## 8. Clinical Implications

Genotyping pancreatic cancer may be a good therapeutic guide, and we may theorize somebasic rules:Low hEnt1/2 expression patients should not be treated with gemcitabine. A FOLFIRINOX protocolcan achieve better results;High RRM1 expression requires the association of an RRM1 inhibitor such as hydroxyurea;Patients with germline mutations of DNA repair genes, such as BRCA1/2, obtain poor benefits with gemcitabine, and in this case, a FOLFIRINOX treatment associated with PARP inhibitors would be better suited. An alternative would be a PARP inhibitor associated with a single alkylating agent, such as cisplatin;Low hCTN1 expression can be improved with proteosomal inhibitors [667].

Gemcitabine resistance treatment should be probed with hydroxyurea and proteasomal inhibitors in association.This scheme has never been tested in clinical trials.

## 9. Conclusions

Chemotherapy and radiation therapy play an important role in pancreatic cancer treatment. However they have not shown a significant impact on progression-free survival or overall survival (OS), in spite of the more than 3000 clinical trials with different drugs and treatment methods. Gemcitabine, the first-line treatment drug, introduced a minimal improvement of OS which is in the range of weeks rather than months. Very meager results for such a fatal disease. The main cause of failure has been gemcitabine resistance.

After 25 years of clinical experience with gemcitabine one thing is clear: it cannot do the job alone. This explains why, in this quarter of century, no significant improvements have been achieved.

At present, gemcitabine has shown some benefits in the neoadjuvant and post-ablation contexts, which represent only 15% of the population with pancreatic cancer. In advanced inoperable tumors (85%), these benefits are close to negligible. Unless solutions to chemoresistance and drug delivery to the cell are found, the situation will not change. This means that gemcitabine has to be associated with other drugs in order to achievebetter results.

PDAC treatment should be multidirectional. This means that many of the factors listed above such as stromal reaction, IL-6, and cytokines in generalshould be also targeted. A multitargeted approach would reduce chemoresistance and increase gemcitabine’s cytotoxic effects.

Gemcitabine is the gold standard therapy for non-resectable PDAC, neoadjuvantschemes, and post-resection treatments. However, response rates in advanced tumors are low and overall survival has improved only slightly. Furthermore, the resistance rate is high and represents a serious limiting factor. Nanoparticle delivery of the drug seems to improve the consistently poor results, but more research is required on the issue before it can become part of bedside medicine.

Desmoplastic stroma are one of the key elements in treatment failure, and there is no adequate scheme to deal with the problem as yet. On the other hand, gemcitabine has become an important tool in neoadjuvant therapy allowing many borderline tumors to be operated on successfully. hENT1 should be considered a marker for possible response to gemcitabine. Finding low expression of this protein should lead the oncologist to use alternate treatments such as the FOLFIRINOX scheme. Systematic verapamil use should be considered as an adjunct to gemcitabine because more than 70% of PDACs express MDR proteins, even when no previous chemotherapeutic drug was administered. Finally, there is no doubt in our mind that gemcitabine needs to be complemented with other drugs, whether those considered in this paper, or any other that may be developed in the future.

Incorporating hydroxychloroquine or other anti-autophagy drugs associated with gemcitabine or gemcitabine-nab-placlitaxel in the neoadjuvant setting seems to be a promising step in the right direction.

PARP inhibitors maypossibly antagonize gemcitabine; however this has not been experimentally proved. Finally, we proposethat hydroxyurea and proteasomal inhibitors should be tested in the context of gemcitabine resistance.

## 10. Future Perspectives

Whether using gemcitabine as a stand-alone treatment or associated with nab-paclitaxel or cisplatin, its pharmaceutical form is a capital issue for achieving good intracellular delivery. In this regard chemodrug-loaded nanoparticles will make the difference. These particles may also contain verapamil and/or other modulators of MDR in addition to the cytotoxic drug. Additionally, other substances including desmoplasia downregulators, extracellular pH modulators, cytokine inhibitors, and histone deacetylase inhibitors, proteasomal inhibitors, and hydroxyureadrugs will complement the treatment.

In this regard, novel therapeutic approaches are being actively investigated. LW6, the inhibitor of HIFs, seems an interesting drug that needs clinical trials before establishing its place in thebattle against gemcitabine resistance. Systematically dealing with exosome production and release may also play a role. The spectrum of future options is wide and they go fromchemical changes on the gemcitabine molecule to new pharmaceutical forms with better access to the tumor, and drugs with a different target.

## Figures and Tables

**Figure 2 cancers-14-02486-f002:**
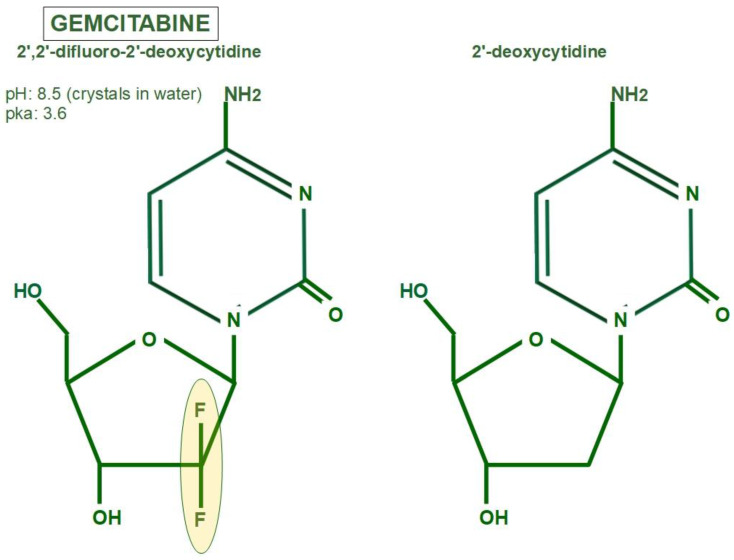
Gemcitabine chemical formula [222] on the left side. The right side shows 2-deoxycytidine (cytosine deoxyribonucleoside), the nucleosidewhich gemcitabine competes against. Cytosine deoxyribonucleoside is one of the four nucleosides that form part of DNA.

**Figure 3 cancers-14-02486-f003:**
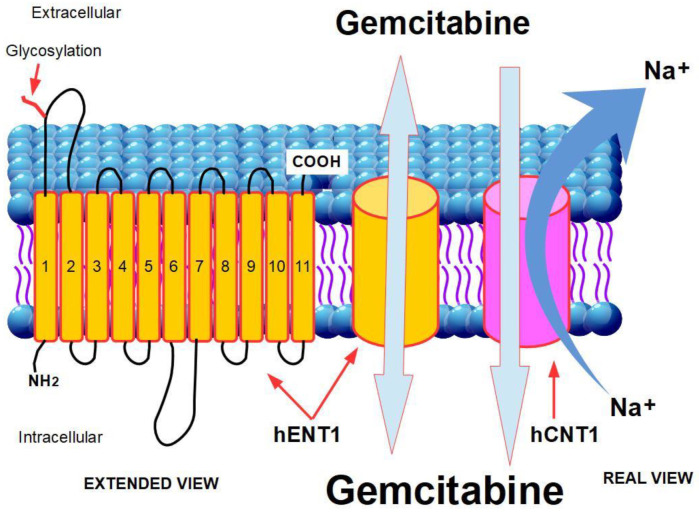
Mechanism of gemcitabine’s access to the cell. Gemcitabine membrane transporters.

**Figure 4 cancers-14-02486-f004:**
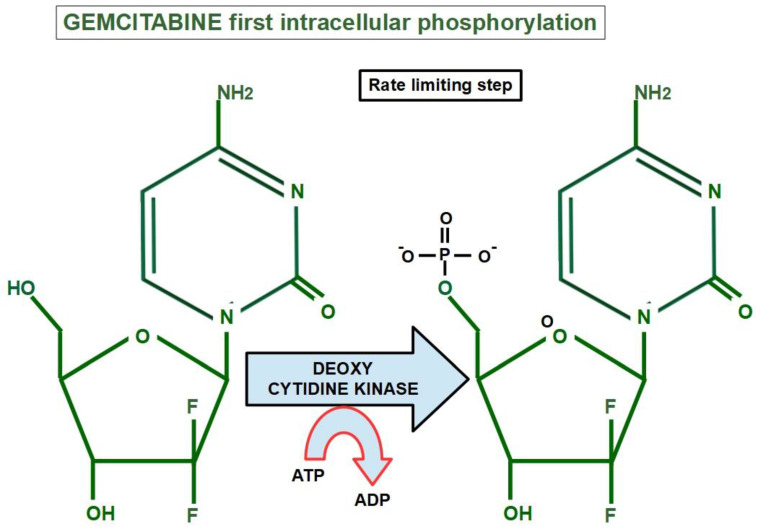
Gemcitabine’s first intracellular phosphorylation by deoxycytidine kinase.

**Figure 5 cancers-14-02486-f005:**
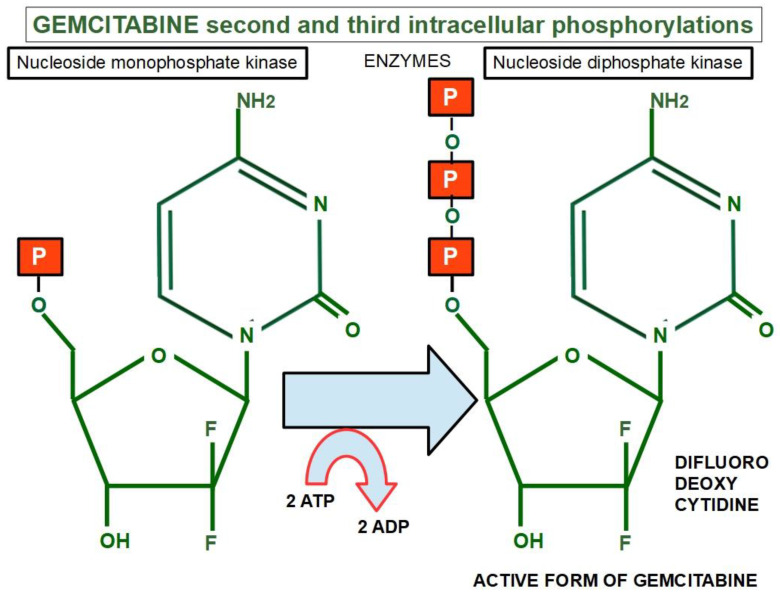
Second and third phosphorylations of gemcitabine by the nucleoside monophosphate kinase and nucleoside diphosphate kinase respectively, rendering the active form: difluoro deoxycytidine.

**Figure 6 cancers-14-02486-f006:**
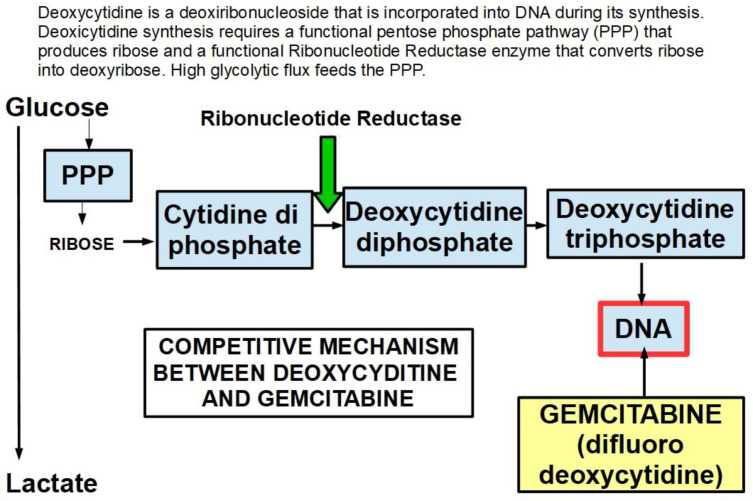
The mechanism of action of gemcitabine is by competing with deoxycytidine. Incorporation of gemcitabine into the DNA strand introduces an irreparable error that the cell cannot circumvent. This faulty DNA unleashes apoptotic mechanisms. A high level of deoxycytidine may prevail over gemcitabine, reducing its effects. The DNA synthesis mechanism is over-simplified in the diagram, the objective of which is to show how an increased glycolytic flux participates in resistance to gemcitabine. Lonidamine, which significantly decreases glycolysis, is probably good to associate with gemcitabine to prevent resistance, although this has not been tested. Increased expression of ribonucleotide reductase, specifically the M1 isoform, is also an important participant in resistance.

**Figure 8 cancers-14-02486-f008:**
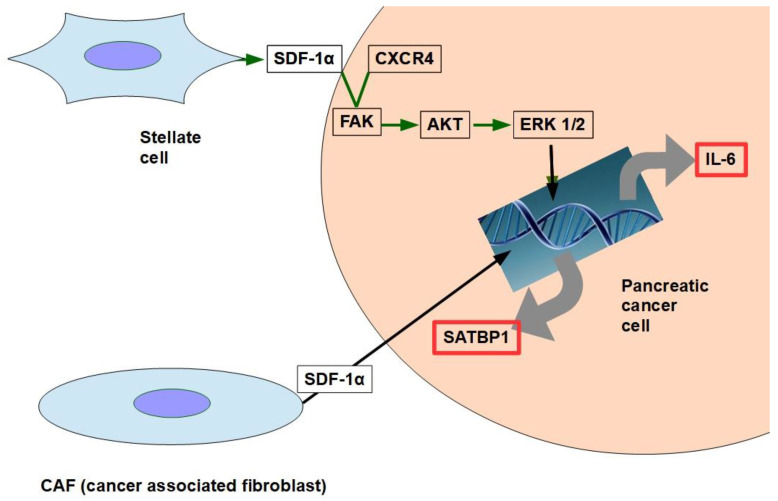
The two pathways shown in the figure have been found to decrease gemcitabine’s cytotoxity and apoptosis. SDF-1α expression is induced by galectin 1.

**Figure 9 cancers-14-02486-f009:**
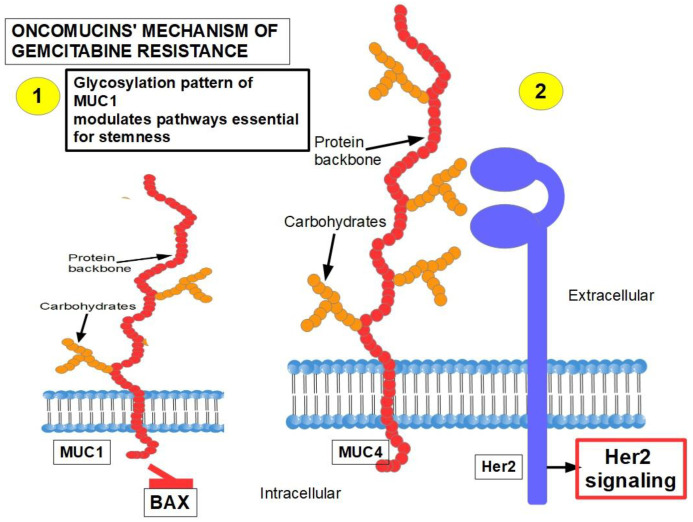
The two mechanisms involved in treatment resistance induced by oncomucins. This diagram is based on references [330,331,332,333,334,335,336,337,338,339,340,341,342,343].

**Figure 10 cancers-14-02486-f010:**
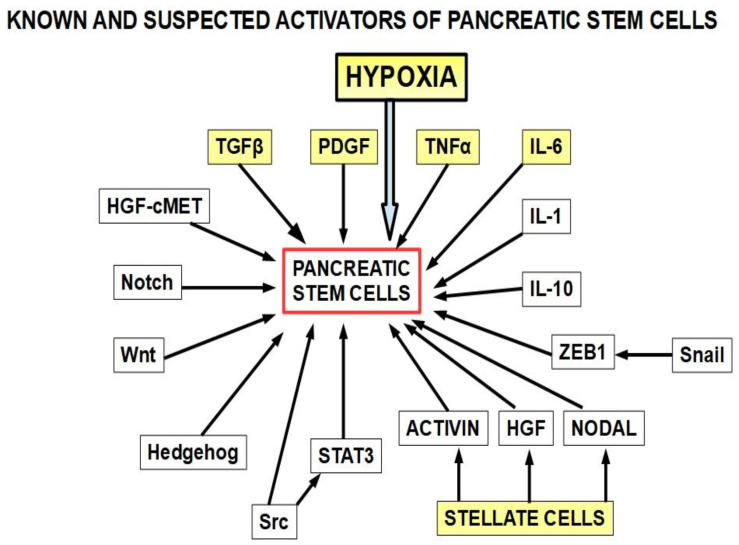
The yellow squares are the known activators of pancreatic cancer stem cells. The other activators (white squares) have also been found to play a role. This diagram is based on references [382,383,384,385,386,387,388,389,390,391,392,393,394,395,396,397,398,399,400,401]. Importantly, many of the stemness activators are also involved in epithelial–mesenchymal transition.

**Figure 11 cancers-14-02486-f011:**
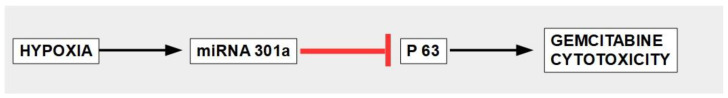
Mechanism of hypoxia-inducedgemcitabine resistance.

**Figure 12 cancers-14-02486-f012:**
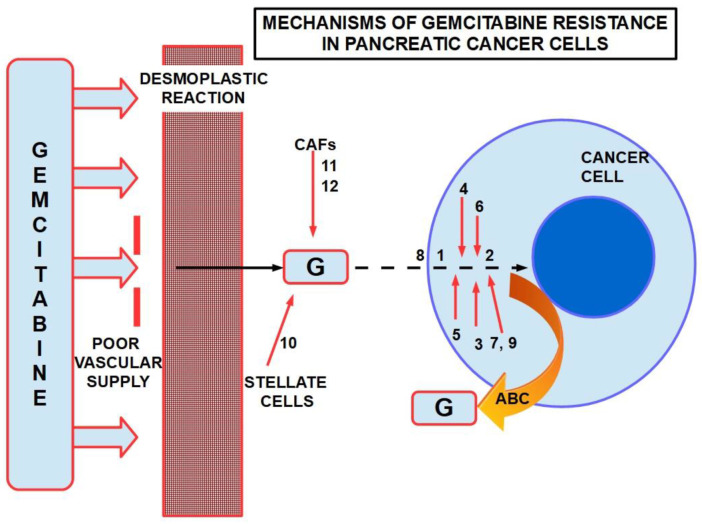
Some mechanisms of resistance to gemcitabine in PDAC. ABC: ATP binding cassette. Poor vascular supply and the desmoplastic reactionare mainly physical barriers. The numbers are chemicals and pathways activated for the escape. ABC re-exports the cytotoxic substances.

**Figure 13 cancers-14-02486-f013:**
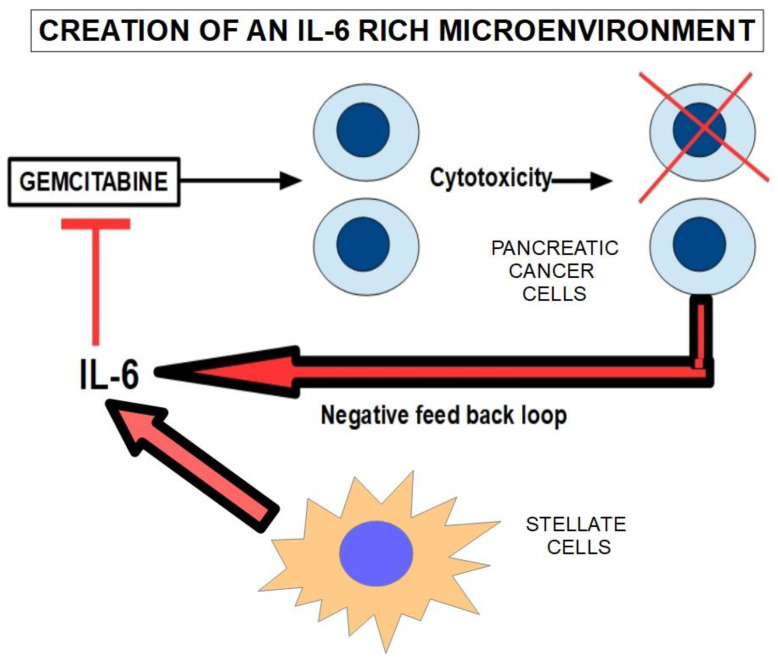
Gemcitabine increases IL-6 expression in the surviving malignant cells, which in turn inhibits gemcitabine’s cytotoxicity through the production/secretion of IL-6.

**Figure 14 cancers-14-02486-f014:**
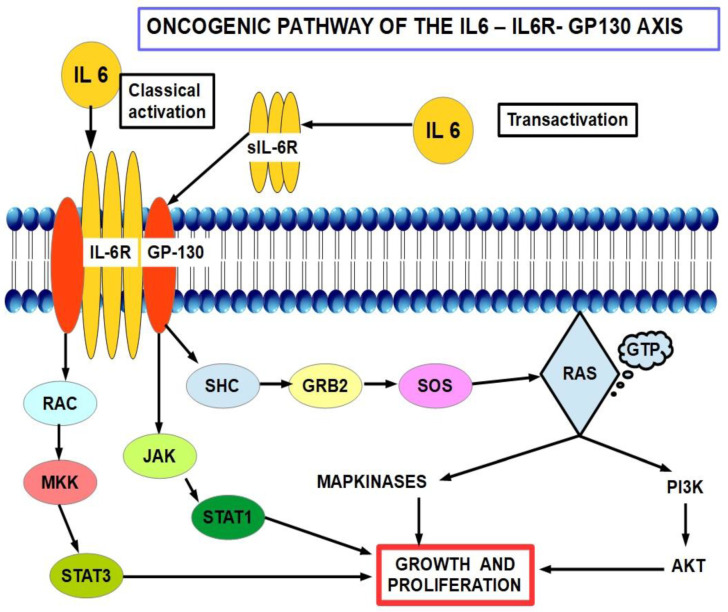
Basedoxifene inhibits GP-130 which is precisely the starting point of IL-6signaling.

**Figure 15 cancers-14-02486-f015:**
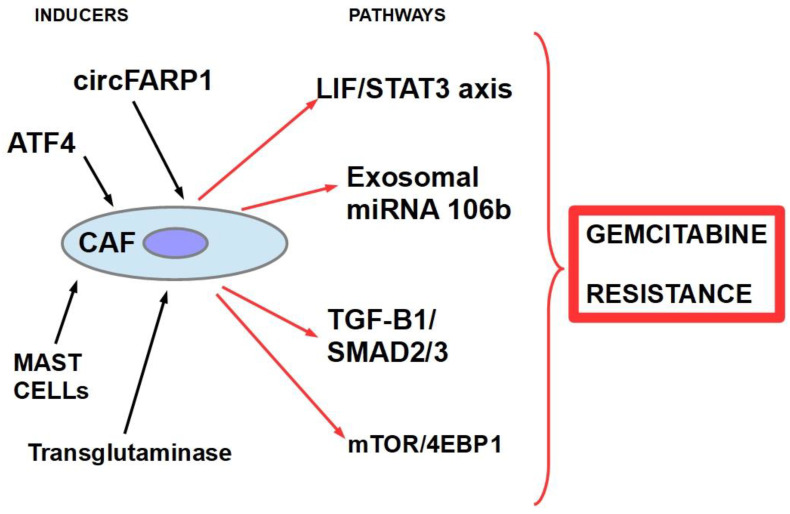
Other pathways that participate in gemcitabine resistance. This diagram is based on references [548,549,550,551,552,553,554,555,556,557,558,559,560].

**Figure 16 cancers-14-02486-f016:**
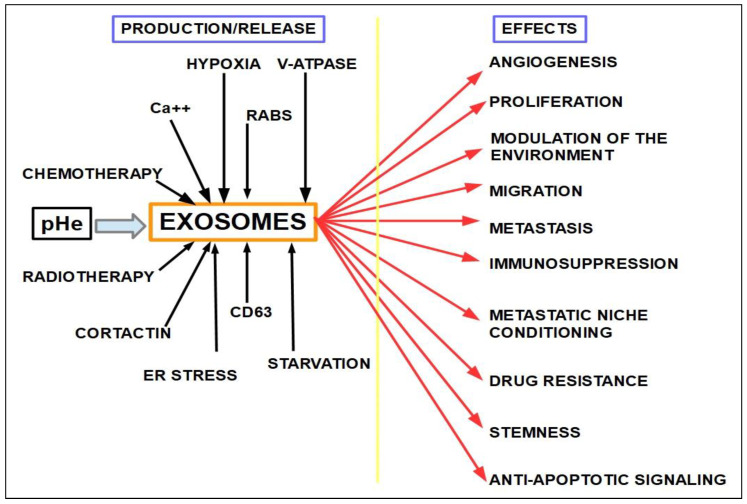
Factors influencing exosome formation and actions. Black arrows show mechanisms that increase exosome production, and red arrows the protumoral effects of these exosomes.

**Figure 17 cancers-14-02486-f017:**
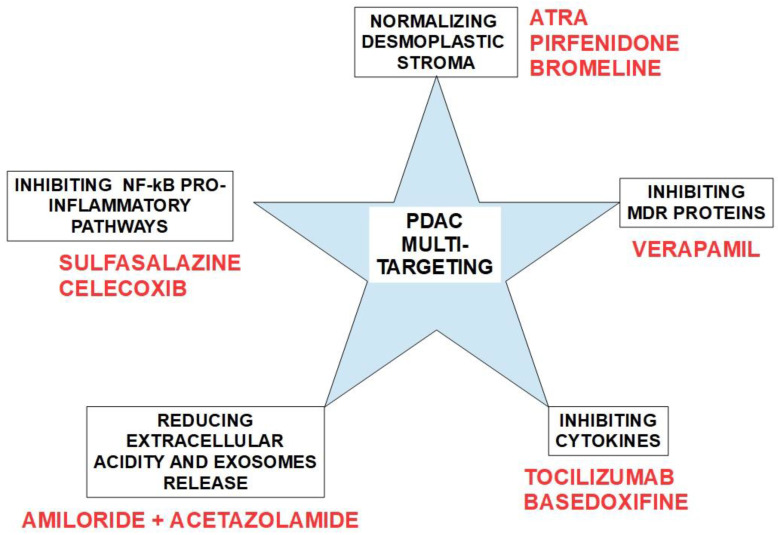
A possible scheme for multi-targeting PDAC to prevent/reverse chemoresistance. A rational association of drugs will probably enhance gemcitabine anti-cancer effects and reduce resistance. The drugs proposed to be associated with gemcitabine have low or no toxicity at all and would not represent an extra burden for the patient. Furthermore, amiloride and tocilizumab may prevent cancer cachexia [603,604], a frequent occurrence in PDAC.

**Figure 18 cancers-14-02486-f018:**
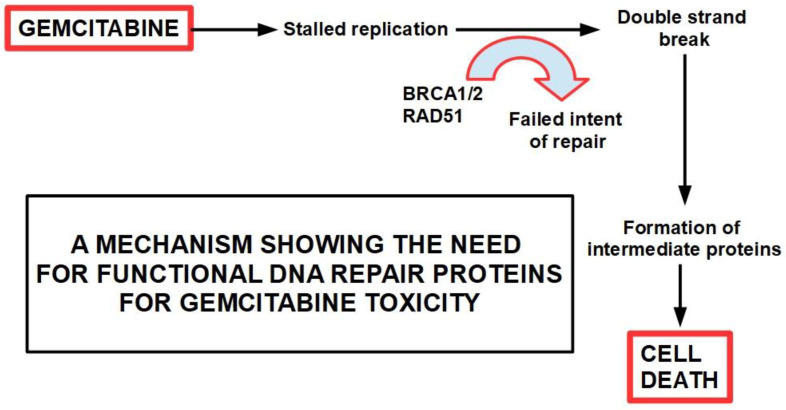
Role of DNA double-strand break-repair proteins in gemcitabine’s cytotoxicity.

**Table 1 cancers-14-02486-t001:** Role of hENT1 expression on effectiveness of Gemcitabine and 5-FU/folinic-acid adjuvant therapy on overall survival in PDAC patients (Greenhalf et al.) [226].

Treatment	High hENT1 (OS)	Low hENT1 (OS)
Gemcitabine	26.2 months	17.1 months
5 FU/folinic acid	21.9 months	25.6 months

**Table 2 cancers-14-02486-t002:** Activity of metabolic intermediaries of gemcitabine.

Gemcitabine (Difluoro Deoxycytidine)	Inactive
Gemcitabine (Difluoro Deoxycytidine Monophosphate)	Inactive
Gemcitabine (Difluoro Deoxycytidine DiphosphateI)	Inhibits RR
Gemcitabine (Difluoro Deoxycytidine Triphosphate)	Inhibits DNA synthesis

**Table 3 cancers-14-02486-t003:** Drugs associated with gemcitabine in clinical trials.

Drug	Mechanism of Action	NCT
Flicatuzumab	Anti-HGF IgG1 mAb	NCT03316599
Durvalumab	Anti-PD-L1 mAb	NCT03572400
Paricalcitol	Vitamin D effects	NCT03520790
CPI-613 (Devimistat)	Targets the pyruvate dehydrogenase complex	NCT03435289
Napabucasin	STAT3 inhibitor	NCT03721744
Gimatecan	Topoisomerase I inhibitor	NCT04571489
SHR-1210	Anti-PD1 mAb	NCT04181645
Abraxane	Protein bound paclitaxel with similar effects to paclitaxel	NCT01693276
Ubidicarenone (BPM- 31510) [622]	Anti-Warburg switch in cancer cell metabolism and activation of apoptosis	NCT02650804
Fruquintinib	VEGFR1, 2, and 3 inhibitor	NCT05168527
Pamrevlumab [623]	Anti-connective tissue growth factor (CTGF) mAb	NCT03941093
OMP-54F28 Ipafricept [624]	Decoy receptor for Wnt ligands. Anti-cancer stem cell	NCT02050178
Vantictumab	Wnt signaling inhibitor mAb, by targeting frizzled receptors	NCT02005315
Erlotinib	EGFR inhibitor	NCT01505413
iNeo-Vac-P01 [625]	Personalized neo-antigen vaccine	NCT03645148
LSL-161 [626]	IAP inhibitor	NCT01934634
MK0752	Notch signaling inhibitor	NCT01098344
Gefitinib	EGFR inhibitor	NCT00234416
Camrelizumab	Immune checkpoint inhibitor	NCT04498689
PBP 1510 [627]	Monoclonal antibody directed against the expression of the oncogene PAUF	NCT05141149
RAV12 [628]	mAb that recognizes an N-linked carbohydrate antigen (RAAG12) expressedin some tumors	NCT00625586
Bevacizumab	Anti-angiogenesis	NCT00460174
Celecoxib	Cox 2 inhibitor	NCT00068432
Enzalutamide	Non-steroidal antiandrogen	NCT02138383
Masitinib	Tyrosine kinase inhibitor anti PDGFR	NCT03766295
Conatumumab	Monoclonal agonist antibody directed against the extracellular domain of TRAIL (tumor necrosis factor-related apoptosis-inducing ligand) receptor 2	NCT01017822
Bey 1107	CDK1 inhibitor	NCT03579836
MM-141	Bispecific antibody against Erb B3 and IGF-IR	NCT02399137
VX-671	Serine proteinase inhibitor	NCT00499265
Nivolumab	Immune checkpoint inhibitor	NCT04247165
S-1 [629]	S-1 consists of three pharmacological agents -Tegafur, a prodrug of 5-FU; 5-Chloro-2-4-Dihydroxypyridine (CDHP), which inhibits the activity of Dihydropyrimidine Dehydrogenase (DPD); and Oxonic Acid (Oxo), which reduces gastrointestinal toxicity of 5-FU	NCT00429858
Lapatinib	EGFR inhibitor	NCT00447122
ABTL0812 [630]	Increases cellular long-chain dihydroceramides which results in sustained ER stress and induces cytotoxic autophagy	NCT03417921
Mirtazapine	Antidepressant	NCT01598584
Simvastatin	Mevalonate pathway inhibitor	NCT00944463
Etoposid	Topoisomerase II inhibitor	NCT00202800
AZD0530 (saracatinib)	Src-Abl inhibitor	NCT00265876
Methyl bardoxolone	Activator of the Nrf2 pathway and an inhibitor of the NF-κB pathway	NCT00529113
Tislelizumab	Anti-PD1 monoclonal antibody	NCT04902261
TBI 1401	Spontaneously attenuated mutant of herpes simplex virus type 1	NCT03252808
ARQ-761	Β-lapachone analogue that causes massive oxidative stress	NCT02514031
GV-1001	Vaccine used foractive immunotherapy of cancers expressing telomerase	NCT02854072
Triapine [631,632,633]	Ribonucleotide reductase inhibitor	NCT00064051
Hua-Chan-Su	Traditional Chinese medicine extract from parotid gland of bufo toads	NCT00837239
Tacedinaline	Histone deacetylase inhibitor	NCT00004861
Icotinib	EGFR tyrosine kinase inhibitor	NCT02278458
Imatinib	Tyrosine kinase inhibitor	NCT00161213
Cabozantinib	Inhibitor of the tyrosine kinases c-Met and VEGFR2	NCT01663272
Ganitumab [634] (AMG 479)	Human monoclonal antibody against IGF-IR	NCT01298401 NCT01318642
IMPRIME PGG BTH1704 [635]	PGG beta glucan is a soluble glucan that binds complement CR3 priming immune cells against opsonized tumor cells. BTH1704 is an Anti MUC1 antibody that binds modified extracellular MUC1	NCT02132403
SLC 0111 [636]	Carbonic anhydrase IX selective inhibitor	NCT03450018
SRF 617 [637]	Blocks CD39 preventing ATP degradation in the extracellular matrix, reducing adenosine and converting an immunosuppressive TME to a proinflammatory environment	NCT04336098
Tislelizumab	PD-1 inhibitor	NCT04902261

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
