# Peer review of "Resistance to Gemcitabine in Pancreatic Ductal Adenocarcinoma: A Physiopathologic and Pharmacologic Review"

_cancers, 2022, doi:10.3390/cancers14102486_

Round 1

Reviewer 1 Report

In this review, the authors provided a comprehensive review of the causes of chemotherapy resistance, especially gemcitabine, in pancreatic ductal adenocarcinoma. The manuscript is very well written, good overview and updated studies. I would recommend a minor revision to this review prior to publication.

  1. In line 147, use “cancer-associated fibroblasts” instead of “cancer-related fibroblasts”.
  2. In line 287, delete “, and,”.
  3. There is no appropriate reference between line 344 and line 347.
  4. The inconsistent description of the most important mechanisms of gemcitabine resistance in pancreatic cancer in Page21, line 703-713 and page 25, line 848-858.
  5. Please carefully revise the English language and grammar for typos, such as in page 11, line 365.

Author Response

  1. In line 147, use “cancer-associated fibroblasts” instead of “cancer-related fibroblasts”.

DONE

  1. In line 287, delete “, and,”.

DONE

  1. There is no appropriate reference between line 344 and line 347.

ADDED REF. 130

  1. The inconsistent description of the most important mechanisms of gemcitabine resistance in pancreatic cancer in Page21, line 703-713 and page 25, line 848-858.

ADDED

  1. Please carefully revise the English language and grammar for typos, such as in page 11, line 365.

CORRECTED

Reviewer 2 Report

The review article by Koltai et al entitled “Resistance to gemcitabine in pancreatic ductal adenocarcinoma. 2 A physiopathologic and pharmacologic review” focused on deciphering the potential mechanisms involved in gemcitabine resistance in pancreatic ductal adenocarcinoma (PDAC). Authors perform a thorough analysis of literature to identify the diverse role of various molecules in gemcitabine resistance, such as transporters, mucins, exosomes, and microRNAs. Though the concept of the article is interesting and supported by relevant literature, the collected information still misses reports of some highly relevant molecules w.r.t. PDAC. Some discrepancies need to be fixed during the revision of the manuscript as a minor revision;

  1. First of all, the introduction sections need to be revised as the author mentioned somehow similar points again and again, such as late diagnosis, surgery etc; the introduction section can be trimmed and more crisp, highlighting the importance of collecting the recent information gemcitabine resistance in PDAC.
  2. Following the introduction, before starting Causes of Resistance, it will be good to provide a brief for PDAC initiation and progression and then start with the resistance problem.
  3. Another important issue is the presentation of important reports in bullet points only if the author wants to give bullets points; it is good to present these outcomes in the form of a Table.
  4. The exosomes play an important role in PDAC, so instead of putting the section on exosomes in the discussion, move this section to the main sections.
  5. The collected information for some of the important regulators that play an important role in gemcitabine resistance is superficial, such as oncomucins and metabolism or tumor microenvironment. MUC1 and MUC4 are well-studied mucins in PDAC specifically expressed in PDAC conditions and resistance. Another important mucin is MUC5AC, which I’m not able to see in the collected information. The author should explain these molecules as multiple reports are available showing the role of these mucins in PDAC and gemcitabine resistance.

Author Response

  • First of all, the introduction sections need to be revised as the author mentioned somehow similar points again and again, such as late diagnosis, surgery etc; the introduction section can be trimmed and more crisp, highlighting the importance of collecting the recent information gemcitabine resistance in PDAC.

REDUNDANCIES IN THE INTRODUCTION HAVE BEEN ELIMINATED

  • Following the introduction, before starting Causes of Resistance, it will be good to provide a brief for PDAC initiation and progression and then start with the resistance problem.

A new heading and paragraph have been introduced

  • Initiation and progression of PDAC

Many possible causative factors have been identified as initiators of this tumor, such as, high plasticity of acinar cells (dedifferentiation into pluripotential cells known as acinar-ductal metaplasia), intra-and peri-tumoral inflammation (including acute and chronic pancreatitis), immunosurveillance failure, KRAS mutation, hyperglycemia, highly variable extracellular pH in acid-base transporting epithelia, ROS regulation by TIGAR, exosomes, nicotine, nicotinic acetyl choline receptors, ER stress protein AGR2, autophagy, and many others [56-129]. The large quantity of proposed tumor initiators, led us to believe that many authors include under the concept of initiation many mechanisms that participate in progression rather as real initiators.

Although the precise initiator of PDAC remains elusive the following facts are clearly known:

  1. There are germline and somatic mutations that predispose to PDAC such as KRAS, p53, p16 and SMAD4. [130-150]
  2. KRAS mutation and activation represents a critical factor in initiation [151, 152].
  3. The origin of pancreatica cancer is from acinar cells [153].
  4. Progression from normal cells into invasive ductal adenocarcinoma is the product of multiple mutations. [154]
  5. Inflammation undoubtedly plays a role in initiation and progression.
  6. Progression, on the other hand, is better known than intiation. It has been established that invasive pancreatic adenocarcinoma is the result of clonal evolution of severe ductal dysplasia [155].

,

  • Another important issue is the presentation of important reports in bullet points only if the author wants to give bullets points; it is good to present these outcomes in the form of a Table.

Some bullet forms have been eliminated

  • The exosomes play an important role in PDAC, so instead of putting the section on exosomes in the discussion, move this section to the main sections.

We prefer to keep the exosomes paragraph in the discussion section. The main section is reversed for what the exosomes transport.

  • The collected information for some of the important regulators that play an important role in gemcitabine resistance is superficial, such as oncomucins and metabolism or tumor microenvironment. MUC1 and MUC4 are well-studied mucins in PDAC specifically expressed in PDAC conditions and resistance. Another important mucin is MUC5AC, which I’m not able to see in the collected information. The author should explain these molecules as multiple reports are available showing the role of these mucins in PDAC and gemcitabine resistance.

Oncomucins have been extensively discussed in the paper, including a specific figure 9 showing the mechanisms of action of oncomucins 1 and 4.

MUC5AC is missing and a paragraph is added on this regard.

MUC5AC, a facilitator of migration and invasion, also participates in drug resistance by inhibiting TRAIL death pathways.

Reviewer 3 Report

Koltai et al. performed a review on resistance to gemcitabine in pancreatic ductal adenocarcinoma (PDAC) from the physiopathologic and pharmacologic prospects. It is great to review this topic nowadays because the last well-written review on this topic was published in 2015 (PMID: 2669034). The main concern about this manuscript is that the authors tried to put as much information as they could in one manuscript, but the arrangement of the whole manuscript needs improvement. A good review should present the up-to-date related information in a highly summarized form to help the reader get the points, not just easily putting everything in one manuscript.

There are some other concerns about this manuscript:

1.Interestingly, although the authors put much information about the nab-paclitaxel in the manuscript, the manuscript did not cite the first paper about the clinical trial that demonstrated that nab-Paclitaxel plus gemcitabine increased survival in pancreatic cancer (PMID: 24131140).

2.Line 202 to line 208, the current writing format is unsuitable for a scientific review paper. It is hard to let the reader get what the authors are talking about, especially “step 5: is deposited”.

3.The section “4. Mechanisms of resistance to Gemcitabine” is poorly arranged. It is helpful that the authors had summarized their long list into three main mechanisms (line 703 to line 707). However, before this, the authors listed 53 mechanisms with multiple figures and boxes (lines 434 to 690), making this section hard to follow. Moreover, while the authors had listed 53 mechanisms for the PDAC gemcitabine resistance, the role of Hippo signaling in PDAC gemcitabine resistance was not discussed. On the other hand, the authors talked a lot about the EMT mechanisms but did not mention the role of the ERK-ZEB-1 pathway in this review.

4.Box2 and Box3 are distracting. This manuscript focuses on resistance to gemcitabine in pancreatic ductal adenocarcinoma. It is unnecessary to use such a big likes Box2 to show readers that it is microRNA. The same issue for Box3, which introduces PARP inhibitors, can be put directly into the main body of the manuscript instead of using a big box in the middle of the manuscript.

5.The contents in small boxes are redundant and lack information (Line 499-501, 561, 566, 569) compared with the main body of the manuscript. 

6.The small box in lines 499 to 501 lacks information and is not helpful. The reader still needs to read the content from lines 495 to 498 to determine if gemcitabine upregulates or downregulates miRNA 155 and how miRNA 155 impacts exosomes.

7.The figures are hard to read and need the figure legends. For example, in figure 16, what does that yellow line mean? What are the different meanings between the black arrows and red arrows?

8.Lines 749 to 750, the authors mentioned that “Nintedanib is undergoing clinical trials for many solid tumors, however, we could not find any related to PDAC.”, which is wrong. University of Texas Southwestern Medical Center is recruiting the patients for the project Study of Nintedanib and Chemotherapy for Advanced Pancreatic Cancer (NCT02902484). The last modification to this clinical trial was 27 February 2022.

9.Lines 1049 to 1052, the authors described that table 2 showed the clinical trials. According to the manuscript, table 2 is “Activity of metabolic intermediaries of gemcitabine”. Moreover, it is unnecessary to show this table 2 in the manuscript. Besides lines 374 to 375, there are two “table2”.

Author Response

Koltai et al. performed a review on resistance to gemcitabine in pancreatic ductal adenocarcinoma (PDAC) from the physiopathologic and pharmacologic prospects. It is great to review this topic nowadays because the last well-written review on this topic was published in 2015 (PMID: 2669034). The main concern about this manuscript is that the authors tried to put as much information as they could in one manuscript, but the arrangement of the whole manuscript needs improvement. A good review should present the up-to-date related information in a highly summarized form to help the reader get the points, not just easily putting everything in one manuscript.

There are some other concerns about this manuscript:

1.Interestingly, although the authors put much information about the nab-paclitaxel in the manuscript, the manuscript did not cite the first paper about the clinical trial that demonstrated that nab-Paclitaxel plus gemcitabine increased survival in pancreatic cancer (PMID: 24131140).

THE REFERENCE WAS INCLUDED AS 31

2.Line 202 to line 208, the current writing format is unsuitable for a scientific review paper. It is hard to let the reader get what the authors are talking about, especially “step 5: is deposited”.

THE SENTENCE HAS BEEN CHANGED

3.The section “4. Mechanisms of resistance to Gemcitabine” is poorly arranged. It is helpful that the authors had summarized their long list into three main mechanisms (line 703 to line 707). However, before this, the authors listed 53 mechanisms with multiple figures and boxes (lines 434 to 690), making this section hard to follow. Moreover, while the authors had listed 53 mechanisms for the PDAC gemcitabine resistance, the role of Hippo signaling in PDAC gemcitabine resistance was not discussed. On the other hand, the authors talked a lot about the EMT mechanisms but did not mention the role of the ERK-ZEB-1 pathway in this review.

THE HIPPO PATHWAY WAS ADDED AS ONE OF THE MECHANISMS THAT PARTICIPATE IN GEMCITABINE RESISTANCE

4.Box2 and Box3 are distracting. This manuscript focuses on resistance to gemcitabine in pancreatic ductal adenocarcinoma. It is unnecessary to use such a big likes Box2 to show readers that it is microRNA. The same issue for Box3, which introduces PARP inhibitors, can be put directly into the main body of the manuscript instead of using a big box in the middle of the manuscript.

THESE BOXES HAVE BEEN REMOVED

5.The contents in small boxes are redundant and lack information (Line 499-501, 561, 566, 569) compared with the main body of the manuscript. 

THESE BOXES HAVE BEEN REMOVED

6.The small box in lines 499 to 501 lacks information and is not helpful. The reader still needs to read the content from lines 495 to 498 to determine if gemcitabine upregulates or downregulates miRNA 155 and how miRNA 155 impacts exosomes.

THE BOX HAS BEEN REMOVED

7.The figures are hard to read and need the figure legends. For example, in figure 16, what does that yellow line mean? What are the different meanings between the black arrows and red arrows?

Figure 16 THE LEGEND WAS MODIFIED EXPLAINING THE MEANING OF THE ARROWS.

8.Lines 749 to 750, the authors mentioned that “Nintedanib is undergoing clinical trials for many solid tumors, however, we could not find any related to PDAC.”, which is wrong. University of Texas Southwestern Medical Center is recruiting the patients for the project Study of Nintedanib and Chemotherapy for Advanced Pancreatic Cancer (NCT02902484). The last modification to this clinical trial was 27 February 2022.

THE TRIAL MENTIONED BY THE REVIEWER IS ONLY FOR NINTEDANIB BUT IT IS NOT ASSOCIATED WITH GEMCITABINE. However we included it in the paper.

9.Lines 1049 to 1052, the authors described that table 2 showed the clinical trials. According to the manuscript, table 2 is “Activity of metabolic intermediaries of gemcitabine”. Moreover, it is unnecessary to show this table 2 in the manuscript. Besides lines 374 to 375, there are two “table2”.

It has been corrected

Round 2

Reviewer 3 Report

For the NCT02902484 clinical trial, here is the summary from the NCT website (https://clinicaltrials.gov/ct2/show/NCT02902484) :

“Brief Summary:

The study will perform a clinical study evaluating the safety and tolerability of nintedanib when combined with standard chemotherapy (Gemcitabine + nab-Paclitaxel) for metastatic pancreatic cancer. It will utilize advanced imaging correlates including dynamic contrast enhanced Magnetic Resonance Imaging (DCE-MRI) which correlates with tumor grade and microvessel density.”

This summary clearly described this clinical trial, which is for nintedanib combined with Gemcitabine + nab-Paclitaxel in PDAC. Why do the authors still claim that “THE TRIAL MENTIONED BY THE REVIEWER IS ONLY FOR NINTEDANIB BUT IT IS NOT ASSOCIATED WITH GEMCITABINE.”?

Author Response

Thank you for establishing our error.

It has been corrected.